# SynHLMA: Synthesizing Hand Language Manipulation for Articulated Object with Discrete Human Object Interaction Representation

## Abstract

Generating hand grasps with language instructions is a widely studied topic that benefits from embodied AI and VR/AR applications. While transferring into **H**and **A**rticulated **O**bject **I**nteraction (**HAOI**), the hand grasps synthesis requires not only object functionality but also long-term manipulation sequence along the object deformation. This paper proposes a novel HAOI sequence generation framework **SynHLMA**, to **Syn**thesize **H**and **L**anguage **M**anipulation for Articulated objects. Given a complete point cloud of an articulated object, we utilize a discrete HAOI representation to model each hand object interaction frame. Along with the natural language embeddings, the representations are trained by an HAOI Manipulation Language Model to align the grasping process with its language description in a shared representation space. An **articulation-aware loss** is employed to ensure hand grasps follow the dynamic variations of articulated object joints. In this way, our SynHLMA achieves three typical hand manipulation tasks for articulated objects of HAOI generation, HAOI prediction and HAOI interpolation. We evaluate SynHLMA on our built HAOI-lang dataset and experimental results demonstrate the superior hand grasp sequence generation performance comparing with state-of-the-art. We also show a robotics grasp application that enables dexterous grasps execution from imitation learning using the manipulation sequence provided by our SynHLMA. Our codes and datasets will be made publicly available.

## 1 Introduction

Human-object interaction is widely studied, yet inferring human manipulation intent remains challenging—especially for enabling robots to learn dexterous manipulation through object functionality understanding and action execution. Current works primarily synthesize rigid-object hand-object interactions (HOI) with physically realistic grasp poses. However, articulated objects demand modeling complete deformation processes beyond simple grasps. For instance, scissor usage requires synthesizing both grasp poses and subsequent manipulation sequences (e.g., opening/closing motions). Thus, learning to generate **H**uman **A**rticulated **O**bject **I**nteractions (**HAOI**) under human intent is critical for robotic dexterity.

Existing grasp synthesis approaches face significant limitations. Methods using robotic hands (Jin et al., 2024; Xie et al., 2023) lack human-hand realism, while skeleton-driven techniques (Yang et al., 2022a; Karunratanakul et al., 2021) ignore hand-object contact physics. Contact prediction models (Grady et al., 2021; Yang et al., 2022b) struggle to integrate language descriptions with articulated object dynamics. More critically, diffusion models exhibit poor long-sequence generation due to insufficient priors, and most methods neglect model generalization beyond text-to-HOI tasks.

To address the aforementioned issues, in this paper, we propose **SynHLMA**, a novel HAOI sequence generation framework to **Syn**thesize **H**and **L**anguage **M**anipulation for Articulated objects. Given an articulated object's point cloud and language query, SynHLMA leverages VQ-VAE to discretize per-frame HAOI representations. These representations are aligned with language embeddings via

LoRA-trained manipulation language models in a shared semantic space. An autoregressive strategy predicting incremental differences addresses long-sequence generation, enabling HAOI generation, prediction, and interpolation (Fig. 1).

Inspired by language's inherent discreteness and grasp taxonomies (Feix et al., 2015), we design discrete HAOI representations with separate codebooks encoding hand position, pose, refinement, and object configuration. During manipulation generation, semantic HAOI tokens are assigned per frame, supervised by our novel *articulation-aware loss* enforcing: hand-object penetration avoidance, joint-configuration accuracy, and pose consistency. The articulation-aware loss consists of three key components: 1) **HAOI Penetration Loss**: During generation, the hand may penetrate articulated objects with varying shapes. This loss constrains the physical contact distance between the hand and objects, ensuring physically plausible interactions. 2) **Joint-Aware Loss**: In practice, the state of articulated joints critically influences hand poses. We incorporate joint-state predictions as a constraint to guarantee accurate perception and representation of joint configurations. 3) **Pose Consistency Loss**: Temporal coherence across consecutive frames is a decisive factor for the quality of sequences generated by the Manipulation Language Model. We introduce a consistency constraint on hand-object transformations to maintain smooth and physically consistent motion across frames. Notably, these three components are distributed across our two-stage training pipeline ensuring that both low-level articulation fidelity and high-level semantic generation are jointly optimized.

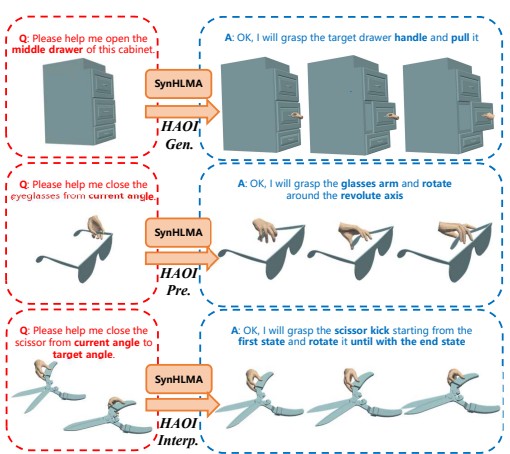

Figure 1: Given an articulated object shape and a natural language as instruction, our SynHLMA achieves three typical hand manipulation synthesis tasks: HAOI generation (up), HAOI prediction (middle) and HAOI interpolation (down) with their corresponding manipulation descriptions.

Accompanying the HAOI synthesis task, we introduce a task-specific dataset, **HAOI-Lang**. Leveraging a physics-based interaction engine, we sample 200 contact points and directions per articulated object to generate extensive HAOI interaction sequences. Following an initial manual filtering, GPT-4 is employed to produce diverse natural language annotations for each interaction, encompassing grasping intent, direction, and location. We evaluate SynHLMA on three tasks: HAOI generation, prediction, and interpolation. Experimental results demonstrate that SynHLMA outperforms existing hand-manipulation generation baselines. Furthermore, we show that the HAOI sequences generated by SynHLMA can effectively guide imitation learning for dexterous robotic hands. In summary, our work makes the following key contributions:

1) **HAOI Language Dataset**: We construct a novel dataset for articulated object grasping, which includes detailed natural language descriptions of grasp intents and actions. 2) **Discrete Manipulation Learning**: We discretize manipulation trajectories using hierarchical grasp tokens, improving generation quality and control. 3) **Articulation-Aware Loss**: By introducing a set of complementary constraints on the HAOI generation process, we mitigate distributional discrepancies among different network components and improve the model's responsiveness to articulated-object variations. 4) **Manipulation Language Model**: We present the first language model for articulated object manipulation, bridging natural language and high-level actions via grasp tokenization to support HAOI generation, prediction, and interpolation.

## 2 RELATED WORKS

### 2.1 HAND GRASP SYNTHESIS

Grasp synthesis for hand poses has a long research history. Following current trends, existing methods can be broadly categorized into two main approaches: probabilistic grasp generation and simulator-based grasp generation.

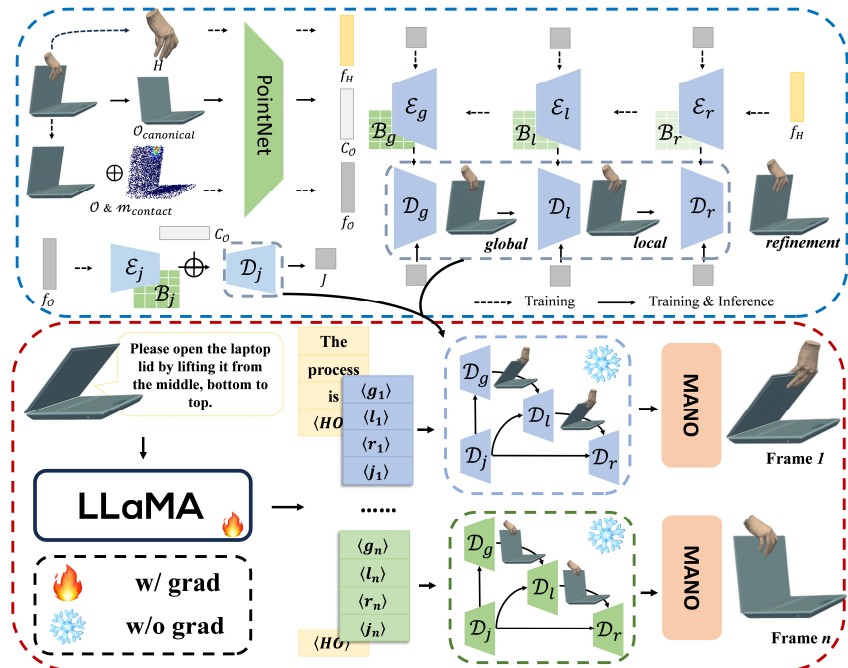

Figure 2: Our SynHLMA pipeline, the upper blue dashed region illustrates the training process of our proposed Discrete Articulated Manipulation Representation model. The lower red region depicts the HAOI Manipulation Language Model, where the parameters of the Discrete Articulated Manipulation Representation are kept frozen during training.

Probabilistic grasp generation can be broadly categorized into two approaches: regression-based and diffusion-based methods. In regression-based models (Li et al., 2024; Huang et al., 2025; Prokudin et al., 2019; Yang et al., 2022b; 2021), a point cloud encoder is first used to extract features representing the HOI. The model then learns the distribution of these features and samples from it, followed by a regression module that predicts the hand parameters to complete the grasp generation. In contrast, diffusion-based methods (Ye et al., 2023; Zhang et al., 2024b) gradually add noise to existing HOI data and learn to reverse this process through denoising. By modeling the denoising trajectory, the model captures the underlying distribution of HOI configurations in the dataset. Grasp generation is then achieved by iteratively denoising a noise-initialized sample.

Although probabilistic generation methods have become increasingly mature and object datasets (Deitke et al., 2023; Mo et al., 2019; Liu et al., 2022) are steadily improving, there remains a significant lack of high-quality HOI datasets. With the advancement of virtual physics engines (Hussain et al., 2020; Hwangbo et al., 2018; Xiang et al., 2020), simulator-based grasp generation has begun to attract growing attention. For example, Xu et al. (2023); Zhang et al. (2024a); Christen et al. (2022) proposed a method that employs a reinforcement learning framework to learn grasping reward strategies, with reward signals provided by a physics engine. This approach enables the large-scale generation of HOI datasets.

## 2.2 MULTIMODEL LARGE LANGUAGE MODELS

In recent years, the remarkable success of large language models (LLMs) such as GPT (Achiam et al., 2023), Qwen (Bai et al., 2023), Gemini (Team et al., 2023) and Gemma (Team et al., 2024) in various downstream natural language tasks—including text translation (Devlin et al., 2019), semantic understanding (Du et al., 2022), and text generation (Zhang et al., 2020) has inspired advancements in other domains. Many researchers have attempted to integrate LLMs with multimodal data, including audio (Borsos et al., 2023), images (Radford et al., 2021), videos (Li et al., 2023), and 3D object models (Jiang et al., 2023), to develop powerful multimodal models.

Jiang et al. (2023); Guo et al. (2022); Huang et al. (2025); Zhang et al. (2023) discretize poses and motions into tokens via encoders, then leverage pretrained large language models (LLMs) to achieve multimodal alignment for motion generation. Cha et al. (2024) aligns object and text features through PointNet and CLIP models respectively, and employs a diffusion model to learn feature distributions for generation. In our work, we adopt a similar paradigm and fine-tune Vicuna to generate grasping sequences conditioned on hand-object interaction features and natural language instructions.

# 3 METHOD

## 3.1 OVERVIEW

We introduce SynHLMA, a framework for generating HAOI sequences. It discretizes full grasping trajectories into token sequences via a multi-stage VQ-VAE with articulation-aware constraints. A manipulation language model learns the distribution of tokenized actions conditioned on language descriptions, enabling complete HAOI sequence generation. We construct a large-scale physics-simulated dataset, with each interaction annotated using GPT.

## 3.2 SYNTHESIS HAOI DATA GENERATION

Existing datasets lack articulated object manipulation sequences paired with language descriptions. A comprehensive joint-body dataset should satisfy: **Completeness** (covering common real-world joint types), **Physical Plausibility** (grasps obey physical and kinematic constraints), **Generality** (including small and large objects), and **Consistency** (object motions and corresponding language are standardized and unambiguous). To address this, we construct **HAOI-Lang**, a novel dataset of HAOI manipulation sequences guided by language instructions, built upon the ArtImage dataset (Xue et al., 2021) within the PartNet-Mobility repository (Xiang et al., 2020), and leverage a physics simulator to emulate realistic manipulation scenarios.

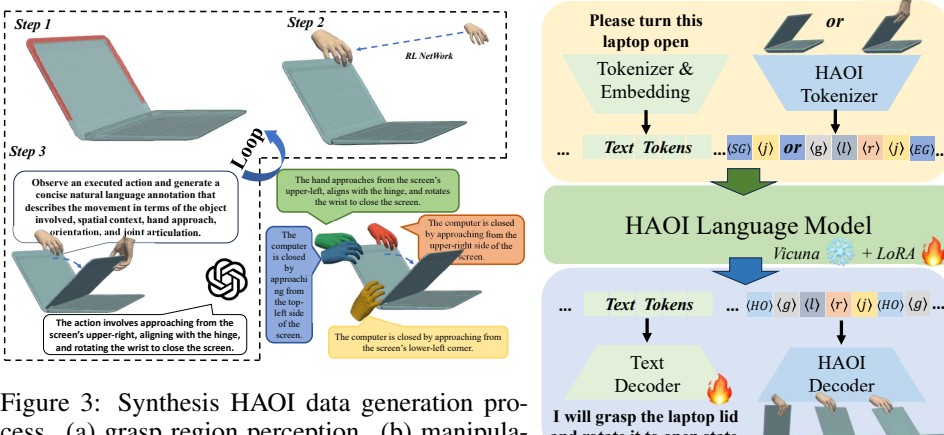

Figure 3: Synthesis HAOI data generation process. (a) grasp region perception. (b) manipulation sequence generation with RL. (c) language instruction generation with GPT-4. (d) large number of samples demonstration.

Figure 4: HAOI Manipulation Language Model

We employ **RaiSim** as the simulation engine, which stands out for its use of the efficient *Articulated Body Algorithm (ABA)* for forward dynamics. By avoiding explicit mass matrix computation and directly solving for generalized accelerations, ABA achieves $\mathcal{O}(n)$ complexity—particularly beneficial for high-DoF systems.

For grasping, we follow the reinforcement learning approach of Zhang et al. (2024a), which unifies hand–object grasp planning across diverse motion objectives, object geometries, and hand morphologies. The method segments objects into graspable and non-graspable regions and uses a reward function $r = r_{\text{goal}} + r_{\text{grasp}}$, with both terms dynamically computed from simulator states to guide hand–object adjustments and optimize grasp execution.

To ensure consistent data generation, we assign each object category a specific manipulation-angle range and uniformly sample grasping frames from it. For any grasp point $p$ relative to the object's center of mass $g_\mathcal{O}$, the grasp direction is $D = p - g_\mathcal{O}$. Each frame provides a homogeneous transformation $\mathcal{T} \in \mathbb{R}^{4 \times 4}$; we normalize coordinates by applying $\mathcal{T}^{-1}\mathbf{x}$, mapping all grasping data into a unified canonical space.

To capture human intent, we render *GraspXL* grasping sequences in *Open3D* and prompt *GPT-4* to produce multi-perspective textual descriptions covering contact points, wrist orientation, and intended action. This ensures concise, consistent annotations (Figure 3). Our dataset includes 256 instances with over 500,000 static grasps and 50,000 manipulation sequences.

## 3.3 DISCRETE ARTICULATED MANIPULATION REPRESENTATION

Similar to previous works, we parameterize the grasp configuration as: $\boldsymbol{\beta} = (R, P, T)$, where $R \in \mathbb{R}^3$ denotes the global rotation of the hand, $P \in \mathbb{R}^{90}$ represents the hand pose, and $T \in \mathbb{R}^3$ denotes the global translation of the hand. We then employ a parametric hand model—specifically, the MANO model—to render the hand mesh $\mathcal{H} \in \mathbb{R}^{778 \times 3}$ based on these parameters: $\mathcal{H} = \mathcal{M}(\boldsymbol{\beta}) = \mathcal{M}(R, P, T)$, where $\mathcal{M}(\cdot)$ denotes the MANO hand model function that maps the grasp parameters to a 3D hand mesh. In this work, to better fit the motion characteristics of the articulated object $\mathcal{O}$ and the hand during the grasping process, we design two separate VQ-VAE discretization frameworks for them, as the image 2.

To ensure consistency between articulated object motions and hand movements, as well as the physical plausibility of grasps, we introduce a joint token $j \in \mathbb{N}$ and train it using a single-stage VQ-VAE. Given an object $\mathcal{O}$ and a grasp contact point $m_{\text{contact}}$, we first extract joint-relevant features using a PointNet encoder. These features are then processed by a joint encoder $\mathcal{E}_j$ and concatenated with the canonical object feature $C_\mathcal{O}$. The combined representation is passed to a joint prediction decoder $\mathcal{D}_j$, which estimates the joint parameters $J \in \mathbb{R}^3$ (e.g., rotation angle or translation).

For the hand reconstruction, to achieve more detailed fitting of its features, we divide the hand generation process into three stages. These stages learn the global pose $\langle \mathbf{g} \rangle$, local articulation $\langle \mathbf{l} \rangle$, and refinement $\langle \mathbf{r} \rangle$ tokens, where $g, l, r \in \mathbb{N}$. We utilize a **multi-stage VQ-VAE** architecture, which consists of three pairs of encoders $\mathcal{D}_i$, decoders $\mathcal{E}_i$, and codebooks $\mathcal{B}_i$, where $i \in \{g, l, r\}$. The hand's point cloud features and the object's features are input into this architecture. After discretization, the features are reconstructed via the corresponding decoders to generate the final grasping configuration.

For the generation of hand configurations, our multi-stage VQ-VAE enables grasp synthesis through a conditional probability framework. First, the global hand information, including global rotation $R$ and translation $T$, is generated conditioned on the global pose token $\langle g \rangle$ and the articulated object joint token $\langle j \rangle$: $\hat{R}, \hat{T} = \mathcal{D}_g(g, j)$. Next, the hand pose $P$ is generated based on the global pose and local articulation token $\langle g, l \rangle$: $\hat{P} = \mathcal{D}_l(g, l, j)$. Finally, in the refinement stage, we obtain the residual offsets $\Delta R, \Delta P, \Delta T$ conditioned on $\langle g, l, r, j \rangle$: $\hat{\Delta R}, \hat{\Delta P}, \hat{\Delta T} = \mathcal{D}_r(g, l, r, j)$. The final predicted hand grasp configuration is then computed as: $\hat{\boldsymbol{\beta}} = (\hat{\Delta R} \cdot \hat{R}, \ \hat{\Delta P} \cdot \hat{P}, \ \hat{\Delta T} + \hat{T})$, where $\hat{\boldsymbol{\beta}}$ denotes the estimated grasp parameters.

A subset of the *articulation-aware loss* is specifically designed to ensure the semantic correctness of the joint-prediction decoder:

**HAOI Penetration Loss.** This term penalizes interpenetration between the predicted hand mesh $\hat{\mathcal{H}}$ and the object surface. Let $V$ denote the vertices of $\hat{\mathcal{H}}$, and $V_{\text{in}} \subseteq V$ be the subset that penetrates the object mesh. For each $v_i \in V_{\text{in}}$, we compute its squared Euclidean distance to the closest point $v_i^\mathcal{O}$ on the object mesh. Averaging over all penetrating vertices mitigates the effect of outliers:

$$\mathcal{L}_P = \frac{1}{|V_{\text{in}}|} \sum_{v_i \in V_{\text{in}}} \left\| v_i - v_i^\mathcal{O} \right\|_2^2 \tag{1}$$

**Joint Aware Loss.** We employ this loss to ensure that the VQ-VAE decoder accurately captures the angular state of the object at each frame. Here, $J_{\text{gt}}$ denotes the ground-truth values.

$$\mathcal{L}_J = \left\| \mathcal{D}_j \left( \left[ \mathcal{E}_j \left( \text{PointNet}(\mathcal{O}, m_{\text{contact}}) \right), C_\mathcal{O} \right] \right) - J_{gt} \right\|_2^2 \tag{2}$$

Combining these two terms, the subset of articulation-aware loss is defined as:

$$\mathcal{L}_{\text{artic}} = \lambda_P \cdot \mathcal{L}_P + \lambda_J \cdot \mathcal{L}_J \qquad (3)$$

In our framework, for the codebooks $\mathcal{B}_j$ and $\mathcal{B}_i$, $i \in \{g, l, r\}$, each feature vector output $Q$ by the encoder is compared to the discrete vectors $q$ in the codebook by computing the squared Euclidean distance. Each feature vector is then replaced with its nearest discrete codebook vector. Assume that each codebook contains $k$ vectors, each of dimension $d$. This quantization process can be formulated as:

$$q_i = \mathcal{E}_i(Q), \quad i \in \{g, l, r, j\}$$
$$\text{index} = \arg \min_k \|q - b_t\|^2, \quad t \in \{0, 1, \ldots, k-1\} \qquad (4)$$
$$q_i = \mathcal{B}_i[\text{index}].$$

VQ-VAE training involves two primary losses: **reconstruction loss**, which drives the decoder to match targets, and **commitment loss**, which keeps encoder outputs close to the codebook while promoting latent-space diversity. The reconstruction loss for the single-stage VQ-VAE applied to articulated objects is already accounted for in Equation 2, so only the commitment loss is computed. The overall loss is obtained by summing all individual terms, formally defined below:

$$\mathcal{L}_r = \lambda_1 \cdot \|\mathcal{M}(R, 0, T) - \mathcal{M}(\hat{R}, 0, \hat{T})\|^2$$
$$+ \lambda_2 \cdot \|\mathcal{M}(R, P, T) - \mathcal{M}(\hat{R}, \hat{P}, \hat{T})\|^2 \qquad (5)$$
$$+ \lambda_3 \cdot \|\mathcal{M}(R, P, T) - \mathcal{M}(\Delta\hat{R} \times \hat{R}, \Delta\hat{P} \times \hat{P}, \Delta\hat{T} + \hat{T})\|^2$$

Meanwhile, the commit loss for tokens at each layer can be expressed by the following:

$$\mathcal{L}_c = \sum_{i \in \{g, l, r, j\}} \|\text{sg}[q_i] - \mathcal{N}_i(q_i)\|^2 + \beta \|q - \text{sg}[\mathcal{N}_i(q_i)]\|^2 \qquad (6)$$

where $\text{sg}[\cdot]$ denotes the stop-gradient operator , $\beta$ is a hyperparameter balancing the commitment cost and $\mathcal{N}_i(\cdot)$ denotes the discretized value in the codebook of token $i$ that is closest to the feature $q_i$. The goal of VQ-VAE is to minimize $\mathcal{L}_{\text{artic}}$, $\mathcal{L}_r$ and $\mathcal{L}_c$ as much as possible.

### 3.4 HAOI MANIPULATION LANGUAGE MODEL

To train a HAOI manipulation language model for grasp perception, we first extract the target object's point cloud using PointNet. As the text primarily describes how to operate the object (e.g., shutting down a computer) including directions and positions, we do not directly provide the contact-point heatmap at this stage. Instead, we obtain the discretized joint token $\langle j \rangle$ using the pre-trained encoder $\mathcal{E}_j$. Each frame's $\langle j \rangle$ token is inserted between the corresponding $\langle SG \rangle$ (Start of Grasp) and $\langle EG \rangle$ (End of Grasp) markers. The resulting sequence is concatenated with the output of the text tokenizer and fed into the model. Depending on the downstream task, the model can also accept the full HAOI representation $\langle g, l, r, j \rangle$, as illustrated in Figure 4.

We represent the manipulation process as a sequence of static frames, each discretized into hierarchical tokens for global motion $\langle g \rangle$, local motion $\langle l \rangle$, refinement $\langle r \rangle$, and object joints $\langle j \rangle$. Thus, each frame is represented as the composite token sequence $\langle g \rangle \langle l \rangle \langle r \rangle \langle j \rangle$, and the full grasping sequence can be expressed as:

$$\langle HO \rangle \langle g_1 \rangle \langle l_1 \rangle \langle r_1 \rangle \langle j_1 \rangle \cdots \langle g_t \rangle \langle l_t \rangle \langle r_t \rangle \langle j_t \rangle \langle HO \rangle,$$

where $\langle HO \rangle$ denotes the start and end of each frame, and $t$ is the total number of steps. For HAOI manipulation generation, $\langle HO \rangle$ tokens are removed, and each tuple $\langle g, l, r, j \rangle$ is sequentially mapped through the pre-trained multi-layer VQ-VAE codebook to obtain latent vectors. These vectors are then decoded by the multi-layer decoder to produce the grasp configuration for each frame.

Multimodal features are projected into a unified semantic space via a linear projection and embedding layer. Denoting all inputs as $S_{\text{input}}$ and outputs as $S_{\text{out}} = \{x_i\}_{i=1}^t \in \mathbb{R}^t$, sequence generation is formulated as a next-token prediction problem:

$$P(x_i | x_{<i}) = P(x_i | x_1, x_2, \ldots, x_{i-1}), i = 1, 2, \ldots, t. \qquad (7)$$

with the objective of minimizing the negative log-likelihood loss:

$$\mathcal{L}_{\text{NLL}} = -\sum_{i=1}^{t} \log P(x_i | x_{<i}) \tag{8}$$

As the final component of our *articulation-aware loss*, we introduce a **pose consistency loss** to constrain the semantic continuity of tokens across adjacent frames generated by the HAOI manipulation language model:

$$\mathcal{L}_C = \begin{cases} \sum_{i=1}^{n} \arccos\left( \frac{\text{Tr}(\Delta\mathcal{R}_{J_i} \cdot \Delta\mathcal{R}_{R_i}^{\top}) - 1}{2} \right) \cdot \frac{180°}{\pi}, & \text{if joint is rotational,} \\ \sum_{i=1}^{n} \|\Delta J_i - \Delta T_i\|_2, & \text{if joint is translational.} \end{cases} \tag{9}$$

Based on the decoding procedure described in Section 3.3, the output tokens of the manipulation language model are converted into the joint axis state $J$ for each frame, as well as the global hand rotation $R$ and translation $T$. For rotational joints, $J$ encodes the object's axis–angle representation. We convert $J$ into a rotation matrix $\mathcal{R}_J$ using the Rodrigues formula, and compute the inter-frame rotational change as $\Delta\mathcal{R}_{J_i} = \mathcal{R}_{J_i}\mathcal{R}_{J_{i-1}}^{\top}$. The hand's rotational change $\Delta\mathcal{R}_{R_i}$ is obtained analogously. For translational joints, $J$ directly represents the object's translation vector; thus, $\Delta J_i$ and $\Delta T_i$ denote the inter-frame displacement of the object and the hand, respectively. Here, $n$ denotes the number of inter-frame transitions, excluding the first frame which has no predecessor.

The HAOI manipulation language model is fine-tuned in two stages: 1) **Multimodal Alignment**: Embed new special tokens and align original embeddings with discrete manipulation representations. 2) **Instruction Tuning**: Freeze Stage 1 embeddings and tokenizer; optimize joint generation of grasp sequences and language outputs for stable, consistent behavior.

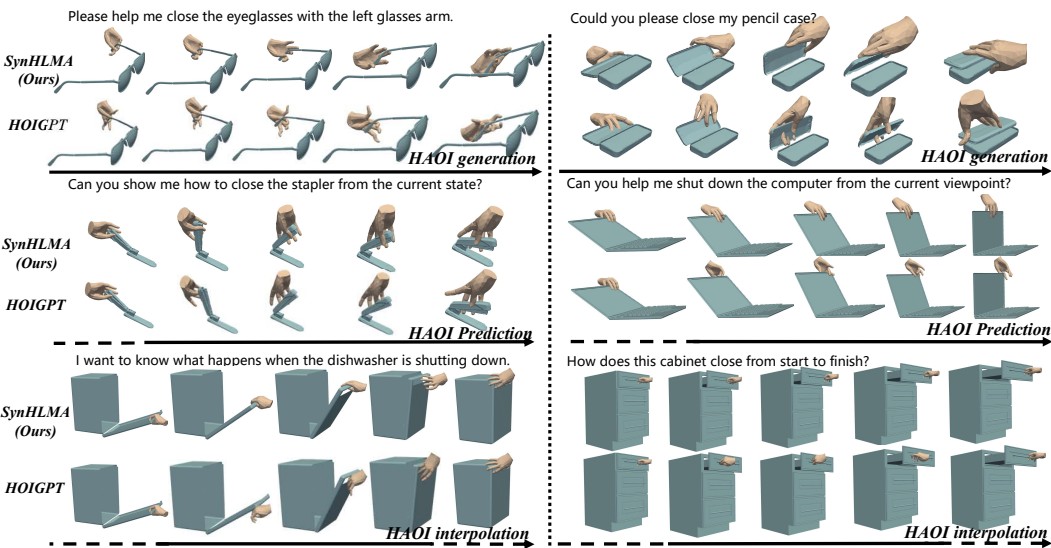

Figure 5: Qualitative results comparison between our SynHLMA and HOIGPT on three HAOI tasks of HAOI generation, HAOI prediction, and HAOI interpolation.

## 4 EXPERIMENTS

### 4.1 EXPERIMENTAL SETTINGS

**Implementation Details** We use four codebooks with a capacity of 1024 each for the hand and the object. For the large language model, we adopt Vicuna-7B-v1.5. The model is trained using the LoRA technique, with a rank value set to 16, meaning about 5% of the parameters are involved in the training process. We set the learning rate to 3e-4 and the alpha value to 64, and employ a cosine annealing learning rate scheduler to ensure training stability. The batch size is 128, and the training is conducted for 20 epochs on four RTX A800 GPUs.

Table 1: Comparison with the state-of-the-art on HAOI generation. The → indicates closer to real is better, and ↓ indicates lower is better.

| Methods | FID ↓ | Diversity → | MMDist ↓ | IV ↓ | ADE ↓ | FDE ↓ |
|---|---|---|---|---|---|---|
| Real | 0.002 | 43.427 | - | - | - | - |
| T2MGPT (Zhang et al., 2023) | 27.027 | 17.124 | 21.007 | 11.218 | 2.145 | 7.013 |
| MotionGPT (Jiang et al., 2023) | 27.361 | 16.876 | 21.625 | 11.436 | 2.119 | 4.879 |
| TM2T (Guo et al., 2022) | 29.459 | 16.968 | 20.964 | 11.975 | 2.014 | 4.856 |
| Text2HOI (Cha et al., 2024) | 22.746 | 21.035 | 23.248 | 13.651 | 1.634 | 2.955 |
| SemGrasp (Li et al., 2024) | 20.873 | 27.954 | 16.482 | 10.247 | 1.284 | 1.932 |
| NL2Contact (Zhang et al., 2024b) | 25.610 | 19.843 | 19.127 | 12.984 | 1.742 | 3.865 |
| HOIGPT (Huang et al., 2025) | 19.040 | 26.498 | 15.003 | 13.828 | 1.055 | 1.168 |
| SynHLMA | **14.121** | **40.484** | **12.793** | **5.919** | **0.976** | **1.147** |

Table 2: Comparison on HAOI prediction and HAOI interpolation.

| Methods | FID ↓ | Diversity → | ADE ↓ | FDE ↓ | FID ↓ | Diversity → | ADE ↓ |
|---|---|---|---|---|---|---|---|
| Real | 0.001 | 48.283 | - | - | 0.001 | 45.465 | - |
| MotionGPT (Jiang et al., 2023) | 45.223 | 21.348 | 1.876 | 3.363 | 46.082 | 20.315 | 1.718 |
| HOIGPT (Huang et al., 2025) | 36.379 | 29.119 | 1.127 | **1.115** | 34.956 | 24.052 | 1.055 |
| SynHLMA | **21.739** | **48.691** | **0.968** | 1.125 | **25.225** | **44.012** | **0.986** |

**Metrics** To comprehensively evaluate the quality of the generated grasping instructions from multiple perspectives, we adopt the evaluation methodology proposed by Jiang et al. (2023); Huang et al. (2025). We consider the following evaluation metrics: Fréchet Inception Distance (FID), Diversity, Multi-modal Distance (MMDist), Interaction Volume (IV), Average Displacement Error (ADE), Final Displacement Error (FDE), Codebook Update Coverage (CUC). For detailed definitions, please refer to the supplementary material.

**Baselines** HAOI generation refers to predicting the subsequent grasping sequence given the object's point cloud and a corresponding textual description as input. HAOI prediction refers to the task where only the first 20% of a grasping sequence is provided to the model, which must then predict the remaining 80%. HAOI interpolation denotes the scenario in which 40–50% of a sequence is intentionally omitted, requiring the model to complete the missing portions in a coherent manner. We compare SynHLMA with state-of-the-art HAOI generation models, including HOIGPT, Text2HOI, SemGrasp and NL2Contact, as well as with representative human motion generation baselines such as T2MGPT, MotionGPT, and TM2M.

### 4.2 COMPARISON WITH STATE-OF-THE-ARTS

As shown in Table 1, our method achieves state-of-the-art performance on the HAOI generation task, outperforming all baselines. Notably, it improves the FID score by **4.919%** and achieves **12.530%** increase in diversity.

Table 2 presents results on the HAOI prediction and interpolation tasks. For HAOI prediction, our method achieves a **14.64%** improvement in FID and outperforms baselines by **19.572%** in diversity. In the interpolation task, we observe a **9.731%** reduction in FID, along with a **19.969%** gain in diversity.

Unlike prior models, our method introduces an articulation-guided decoder to explicitly capture articulated object structure, and employs a hierarchical HAOI token representation for fine-grained manipulation modeling. Finally, the proposed articulation-aware loss guides the model to generate grasps that are more consistent with the articulated object's physical state and kinematic configuration. Qualitative results of the proposed method are illustrated in Figure 5.

### 4.3 ABLATION STUDIES

**Articulation-aware Loss Ablations** In Table 3, we provide a detailed evaluation of the proposed articulation-aware loss designed for articulated objects. The experimental results show that: 1)

Table 3: Ablation on articulation-aware loss.

| | FID ↓ | IV ↓ | ADE ↓ |
|---|---|---|---|
| w/o $\mathcal{L}_{artic}$ | 15.872 | 6.452 | 1.041 |
| w/o $\mathcal{L}_C$ | 14.990 | 6.104 | 1.112 |
| w/o $\mathcal{L}_{artic} \wedge \mathcal{L}_C$ | 16.840 | 6.987 | 1.233 |

Table 4: Ablation on VQ-VAE design.

| | FID ↓ | Diversity ↑ | IV↓ | CUC |
|---|---|---|---|---|
| $\mathcal{B}$ entries $\mathcal{K} = 512$ | 1.237 | 2.944 | 3.690 | 0.232 |
| $\mathcal{B}$ entries $\mathcal{K} = 2048$ | **0.913** | **3.063** | 3.222 | 0.195 |
| $\mathcal{B}$ dim. $d_{\mathcal{B}} = 256$ | 1.113 | 3.704 | 3.446 | - |
| $\mathcal{B}$ dim. $d_{\mathcal{B}} = 1024$ | 1.055 | 3.235 | **3.195** | - |
| EMA + Reset | 3.637 | 3.327 | 3.704 | - |

Table 5: Ablation on discrete representation.

| | FID ↓ | Diversity ↑ | ADE ↓ |
|---|---|---|---|
| $\langle g, l \rangle$ | 0.976 | **3.428** | 0.974 |
| $\langle g, l, r \rangle$ | 1.152 | 3.379 | 0.988 |
| $\langle g, l, r \times 2, j \rangle$ | 1.696 | 2.790 | 0.989 |
| $\langle g, l, r \times 3, j \rangle$ | 1.179 | 2.715 | 0.893 |
| w/o $\langle g, l, r, j \rangle$ | 1.160 | 3.411 | 1.012 |
| w/o semantic | 1.055 | 3.244 | 0.948 |
| Shared Codebook | 0.940 | 3.086 | 0.925 |
| $\langle g, l, r, j \rangle$ (Ours) | **0.699** | 3.109 | **0.815** |

Table 6: Ablation on manipulation language model setting.

| | FID ↓ | MMDist ↓ | IV ↓ |
|---|---|---|---|
| w/ Llama | 51.911 | 17.222 | 9.244 |
| w/ Qwen | 73.590 | 25.272 | 6.848 |
| w/ Gemma | 22.576 | 16.228 | 2.317 |
| LoRA $r = 8$ | 126.954 | 21.020 | 7.703 |
| LoRA $r = 32$ | 53.485 | 18.737 | 7.580 |
| w/o 2-stage | 39.849 | 19.472 | 7.237 |

Without the static-frame constraint term $\mathcal{L}_{\text{static}}$, both IV and FID increase significantly, indicating reduced physical plausibility and contact consistency in HAOI grasping, as well as larger errors in the generated grasp poses. 2) When removing $\mathcal{L}_C$, the ADE rises markedly, suggesting that temporal inconsistencies—such as noticeable "jittering" or "discontinuities" in hand–object angular transitions—become more pronounced. 3) When both loss terms are ablated together, the overall sequence generation quality drops to the worst level. These results demonstrate that the proposed articulation-aware loss effectively links the key components of the model and explicitly encodes both static HAOI states and continuous manipulation dynamics, thereby ensuring high-quality generation.

**Multi-Level VQ-VAE Ablations** We conduct an ablation study on our multi-level VQ-VAE design (Table 4). Experiments 1 through 4 vary the capacity of the codebook and embedding dimensionality, showing that larger capacities and higher dimensions preserve richer representations and consistently improve all metrics. In 5), we compare training strategies for mitigating codebook gradient collapse. While EMA combined with codebook reset has been reported to help, we find EMA yields limited gains for HAOI generation; thus, our main experiments adopt reset alone. Finally, we track the CUC across evaluations by averaging the three codebooks. Notably, a codebook capacity of only 512 achieves a CUC of 23.2%, indicating that even relatively compact discrete representations suffice for our task.

**Token Semantics in Discrete Representations Ablations** We evaluate the impact of token semantics through a series of ablation studies on our stage-wise discrete token design (Table 5). 1) and 2), which remove the object joint token $\langle j \rangle$, result in a significant performance drop and render the refinement token $\langle r \rangle$ ineffective. 3) and 4) introduce additional fine-grained refinement tokens; although they offer slight improvements, they increase training complexity and hinder convergence. In 5), we replace the VQ-VAE with a standard autoencoder to validate the effectiveness of using a discrete representation. In 6), removing all semantic meanings from tokens leads to degraded performance, while 7), which shares a single codebook across all stages, fails to capture the complexity of grasping behaviors.

**Manipulation Language Model Ablations** Our ablation studies on the HAOI manipulation language model are detailed in Table 6. 1) Comparing Llama-7B with instruction-tuned Vicuna-7B reveals the latter's superior adaptability for text-to-manipulation generation. In 2) and 3), we replace the backbone of the HAOI Manipulation Language Model with Qwen-2.5-7B-Instruct and Gemma-7B-IT, respectively. Among the compared models, the Qwen series shows weaker capability in understanding fine-grained action descriptions and predicting values in the corresponding numerical space, resulting in inferior overall performance. For Gemma, we adopt 4-bit quantization during training to fit our computational constraints, and the information loss introduced by quantization may lead to performance that is slightly lower than that of our original backbone. 4) and 5)

Investigating LoRA rank demonstrates that excessively high ranks induce catastrophic forgetting of prior knowledge by increasing trainable parameters; conversely, overly low ranks impede convergence due to underfitting, leading us to select rank 16 as optimal. 6) Compared to two-stage training, the single-stage approach underperforms and exhibits significantly greater instability.

## 4.4 DEXTEROUS MANIPULATION TRANSFER TO ROBOTICS

We demonstrate the application of our SynHLMA in embodied robotics area, and transfer the learned manipulation sequence poses into ShadowHand model Li et al. (2019) within RaiSim simulator in Fig. 6. In detail, we align the finger keypoints from MANO model generated by SynHLMA and ShadowHand model with a simple fitting-based optimization approach. Next, using the fitted ShadowHand poses, we can enable the dexterous manipulation that implements on the same articulated objects.

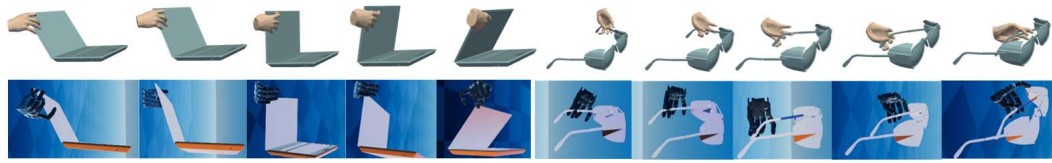

Figure 6: Application of our SynHLMA transferring into ShadowHand model in the robotics scenario

## 5 CONCLUSION

In this work, we propose **SynHLMA**, a novel manipulation language model for articulated objects based on discrete representations. To support this framework, we introduce **HAOI-Lang**, a new dataset that bridges the gap between manipulation actions and their corresponding language instructions. By designing multi-level tokens, our model captures fine-grained grasping processes, achieving strong performance across various downstream tasks. Moreover, an **articulation-aware loss** is incorporated to ensure consistency between the generated hand motions and the articulated object joints. Furthermore, we demonstrate its potential in enhancing dexterous grasping for robotic hands. In future work, we aim to explore more fine-grained and coordinated bimanual manipulation.

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

## A APPENDIX

### A.1 OVERVIEW

In this supplementary material, we first describe in detail the usage of Large Language Models (LLMs) in our paper. We then provide a comprehensive explanation of the composition and generation method of the HAOI-Lan dataset. Next, we present an in-depth description of the experimental setup, including batch loss functions and scoring metrics not covered in the main paper. Finally, we showcase the visualization results of our Discrete Articulated Manipulation Representation across multiple downstream HAOI tasks, demonstrating its generalization ability across different objects within the same category.

### A.2 USE OF LARGE LANGUAGE MODELS

In the preparation of this paper, we used ChatGPT and DeepSeek to assist with language polishing. We also consulted ChatGPT for inspiration regarding the code architecture. All substantive research design, implementation, data analysis, and writing decisions were made by the authors. Every output generated with the help of LLMs was carefully reviewed, verified, and revised by the authors. The authors retain full responsibility for all content in this paper.

### A.3 HAOI-LANG DATASET CONSTRUCTION

#### A.3.1 DATASET COMPOSITION AND OBJECT COVERAGE

Our HAOI-Lan dataset consists of seven commonly encountered articulated objects: **stapler**, **laptop**, **scissors**, **cabinet**, **dishwasher**, **eyeglasses**, and **box**. These objects collectively cover nearly all major types of articulated structures found in real-world environments, such as rotational joints, sliding mechanisms, and compound linkages. Each object category includes around 30–70 object instances, ensuring significant intra-class variation in appearance, articulation, and geometry.

For each object instance, we first segment the graspable regions from the non-graspable parts. Within these graspable regions, we **uniformly sample 200 surface points**. Around each sampled point, we simulate or generate a batch of manipulation sequences, capturing plausible interactions grounded in object geometry and affordance. To enhance semantic richness, we leverage GPT-4 to generate a unique caption for each manipulation episode, describing the intent or action in natural language. The detailed statistics of the dataset are shown in Table 7.

#### A.3.2 SEMANTIC CAPTIONING AND INSTRUCTIONAL DIALOGUE GENERATION

As described in the main paper, we use Open3D to visualize each manipulation episode into a series of sequential image frames. These frame sequences are then passed to ChatGPT-4, which generates a detailed natural language description of the action, capturing the spatial relations, articulation dynamics, and hand-object interactions.

To align with the Instruction-to-Motion task format, we further utilize GPT-4 to transform the detailed captions into instructional dialogues—specifically, concise directives issued from an instructor

Table 7: Statistics of the HAOI-Lang dataset. Each object category contains multiple instances, and each instance includes 200 manipulation episodes paired with 200 unique GPT-4-generated captions.

| Category | Instances | Angle | Manipulations & Captions | Static Grasp |
|----------|-----------|-------|--------------------------|--------------|
| Stapler | 8 | 10 | 1600 | 16000 |
| Box | 8 | 10 | 1600 | 16000 |
| Laptop | 55 | 10 | 11000 | 110000 |
| Scissors | 45 | 10 | 9000 | 90000 |
| Cabinet | 38 | 10 | 7600 | 76000 |
| Dishwasher | 37 | 10 | 7400 | 74000 |
| Eyeglasses | 65 | 10 | 13000 | 130000 |
| **Total** | **256** | - | **51200** | **512000** |

to an executor. We designed a specialized prompt to guide GPT-4 in generating this dialogue format, with the goal of enriching our dataset with task-driven, goal-oriented language instructions.

An example of such a dialogue template is shown in Figure 7.

**System Prompt / Task Definition:**You are an expert in human-robot interaction. Given a detailed description of a manipulation action, your task is to rewrite it as a polite, clear, and concise instruction that an instructor would give to an executor in a collaborative task setting.

**Input (Detailed Description):**"The action involves approaching from the screen's upper-right, aligning with the hinge, and rotating the wrist to close the screen."

**Expected Output (Instructional Command):**"Please approach the screen from the upper-right, align your hand with the hinge, and gently rotate your wrist to close it."

Figure 7: Instructional dialogue template created by GPT-4 for HAOI-Lang.

## A.4 EXPERIMENTS DETAILS

### A.4.1 IMPLEMENTATION DETAILS

**Discrete Articulated Manipulation Representation** For this task, we employ a multi-level VQ-VAE where each codebook is of size 1024x512. There are four codebooks in total, each corresponding to different levels of articulation: global pose, local articulation, and refinement of both hand and object poses. The training configuration for this model includes:

- **Learning Rate:** 0.00002
- **Batch Size:** 128
- **Number of Workers:** 8
- **Number of Points:** 2048
- **Pose Loss:** 1.0
- $\lambda_P$ **:** 1.0
- $\lambda_C$ **:** 1.0
- $\lambda_J$ **:** 1.0

- $\lambda_1$ : 1.0
- $\lambda_2$ : 1.0
- $\lambda_3$ : 1.0
- $\lambda_4$ : 1.0

**HAOI Manipulation Language Model** To equip the model with the ability to understand and generate language instructions for manipulation tasks, we fine-tuned the Vicuna-7B v1.5 model using parameter-efficient adaptation via LoRA. The input to the model consists of manipulation instructions, and the output is a structured semantic representation related to object interaction. The model was trained on high-quality instruction-action paired data with the following key settings:

- **Base Model:** Vicuna-7B
- **Epochs:** 20
- **Batch Size:** 128 (with a micro batch size of 6, using gradient accumulation)
- **Learning Rate:** 3e-4
- **Maximum Input Length:** 256 tokens
- **Validation Set Size:** 11,000
- **LoRA Rank ($r$):** 16
- **Scaling Factor ($\alpha$):** 32
- **Dropout:** 0.05
- **Target Modules:** q_proj, v_proj, embed_tokens, lm_head

### A.4.2 EVALUATION METRICS

To complement the evaluation methodology discussed in the main paper, we provide a detailed account of the quantitative metrics used to assess the quality, diversity, and accuracy of the generated hand-object interaction sequences. These metrics are inspired by previous work in instruction-conditioned motion generation Jiang et al. (2023); Huang et al. (2025) and are tailored to capture both spatial precision and behavioral richness. In addition, to verify that the capacity of the VQ-VAE can adequately discretize continuous tasks, we designed a metric to measure its utilization rate.

**Fréchet Inception Distance (FID)** is used to measure the distributional similarity between the real and generated interaction trajectories. Specifically, it compares the feature embeddings (typically from a pretrained 3D encoder or motion encoder) of both sets using the Fréchet distance:

$$\text{FID} = \|\mu_r - \mu_g\|_2^2 + \text{Tr}(\Sigma_r + \Sigma_g - 2(\Sigma_r \Sigma_g)^{1/2}) \tag{10}$$

where $(\mu_r, \Sigma_r)$ and $(\mu_g, \Sigma_g)$ are the means and covariances of real and generated features, respectively. Lower FID scores indicate higher generation fidelity.

**Diversity** reflects how varied the generated motion trajectories are across different random seeds or conditions. It is calculated as the average pairwise cosine distance between the generated feature vectors:

$$\text{Diversity} = \frac{2}{N(N-1)} \sum_{i<j} \left( 1 - \frac{f_i \cdot f_j}{\|f_i\|\|f_j\|} \right) \tag{11}$$

Higher diversity scores indicate a richer variety of generated behaviors.

**MultiModality (MModality)** evaluates the variability of outputs when the input instruction is held constant. It is computed as the average pairwise distance between generated samples conditioned on the same instruction:

$$\text{MModality} = \frac{2}{K(K-1)} \sum_{i<j} \|x_i - x_j\|_2 \tag{12}$$

where $x_i$ and $x_j$ are generated trajectories under the same condition, and $K$ is the number of generated samples per condition.

**Interaction Volume (IV)** measures how much of the object's surface is covered by the hand during interaction. It quantifies the cumulative 3D space traversed by the hand that is in proximity to the object, typically computed via voxelization:

$$\text{IV} = \sum_{v \in \mathcal{V}} \mathbf{1}\left[\text{dist}(v, H) < \epsilon\right] \tag{13}$$

where $\mathcal{V}$ is the set of object voxels, $H$ is the hand trajectory, and $\epsilon$ is a distance threshold (e.g., $1\,\text{cm}$). The result is reported in $\text{cm}^3$.

**Average Displacement Error (ADE)** quantifies the temporal accuracy of the predicted hand trajectory with respect to the ground truth. It is defined as the mean Euclidean distance across all time steps:

$$\text{ADE} = \frac{1}{T} \sum_{t=1}^{T} \|x_t - \hat{x}_t\|_2 \tag{14}$$

where $x_t$ and $\hat{x}_t$ denote the predicted and ground-truth hand positions at time step $t$.

**Final Displacement Error (FDE)** captures the spatial deviation at the final frame of the trajectory:

$$\text{FDE} = \|x_T - \hat{x}_T\|_2 \tag{15}$$

FDE is especially important for evaluating whether the generated trajectory reaches the intended end state.

**Codebook Update Coverage (CUC)** is used to measure the proportion of codebook entries that are actively updated during each training or evaluation round. Specifically, it quantifies how extensively the discrete latent space of a VQ-VAE is utilized:

$$\text{CUC} = \frac{N_{\text{updated}}}{N_{\text{total}}} \tag{16}$$

where $N_{\text{updated}}$ is the number of codebook entries updated in the current round and $N_{\text{total}}$ is the total number of entries in the codebook. Higher CUC values indicate broader utilization of the codebook and thus better coverage of the latent space.

## A.5 GENERALIZATION ACROSS OBJECT SCALES IN HAOI TASKS

To demonstrate the diversity and generalization ability of our generative model, we visualize three types of downstream tasks—*HAOI generation*, *HAOI prediction*, and *HAOI interpolation*—on four representative articulated objects. These include two small- to medium-sized objects: **glasses** and a **laptop**, and two large-scale objects: a **cabinet** and a **dishwasher**.

The visualizations (see Figure 8, Figure 9, Figure 10 and 11) show that:

- Our model is capable of generating **distinct and plausible HAOI motions at different locations** on the same object, effectively adapting to local geometry.
- Even when the **grasping location is fixed**, the model can produce **diverse grasp directions**, reflecting its ability to represent multimodal interaction possibilities.
- Importantly, the model maintains **robust performance even on large articulated objects**, successfully generating and inferring physically plausible hand-object interactions without degradation in quality.

These results highlight the flexibility of our discrete representation and its effectiveness in modeling a wide range of manipulation behaviors across object scales and task types.

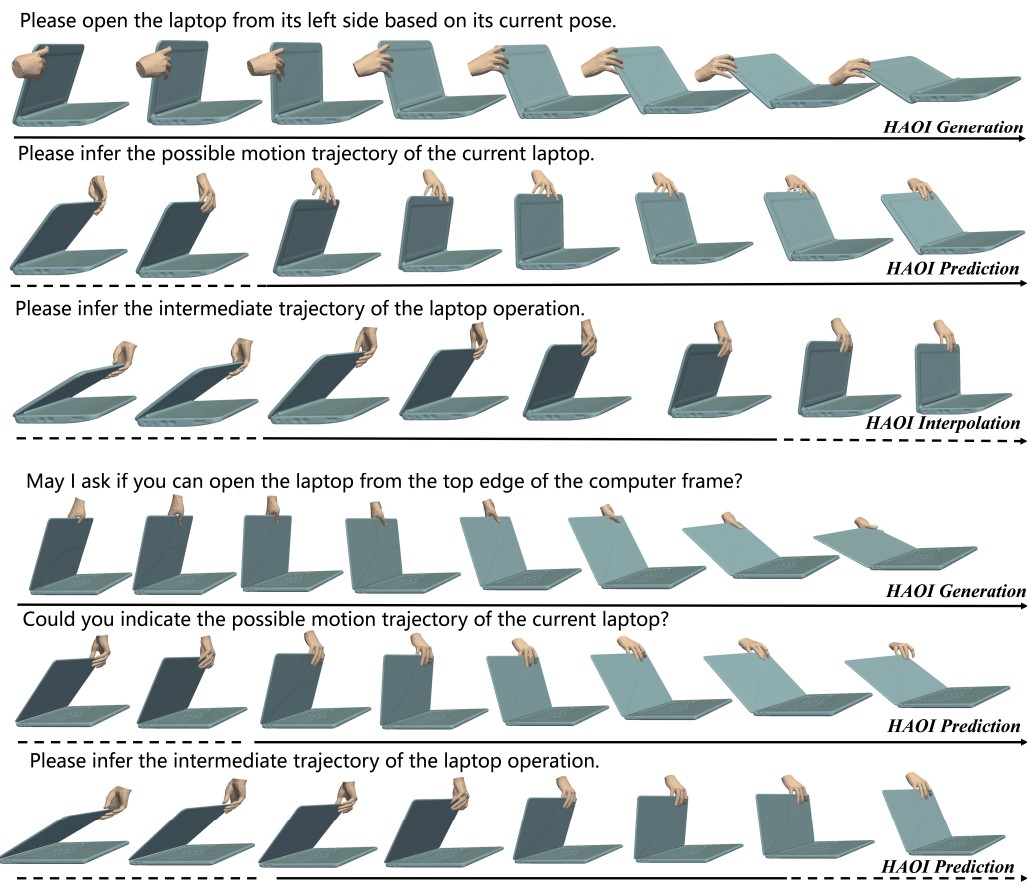

Figure 8: HAOI generation, prediction, and completion on small-to-medium objects: laptop.

## A.6 FAILURE CASES IN HAOI MANIPULATION

We select the most representative failure cases from the test set, as illustrated in Fig. 12. In **HAOI generation**, when the manipulation description is overly strict or physically implausible, the resulting grasp may deviate from realistic or physically feasible configurations. In **HAOI prediction**, the predicted interaction sequence may exhibit temporal inconsistencies, such as hand drift from the intended grasping position or deformation of the grasp over time. In **HAOI interpolation**, the primary issue is abrupt changes in hand pose, which are often caused by discontinuities or invalid predictions in the interpolated pose sequence.

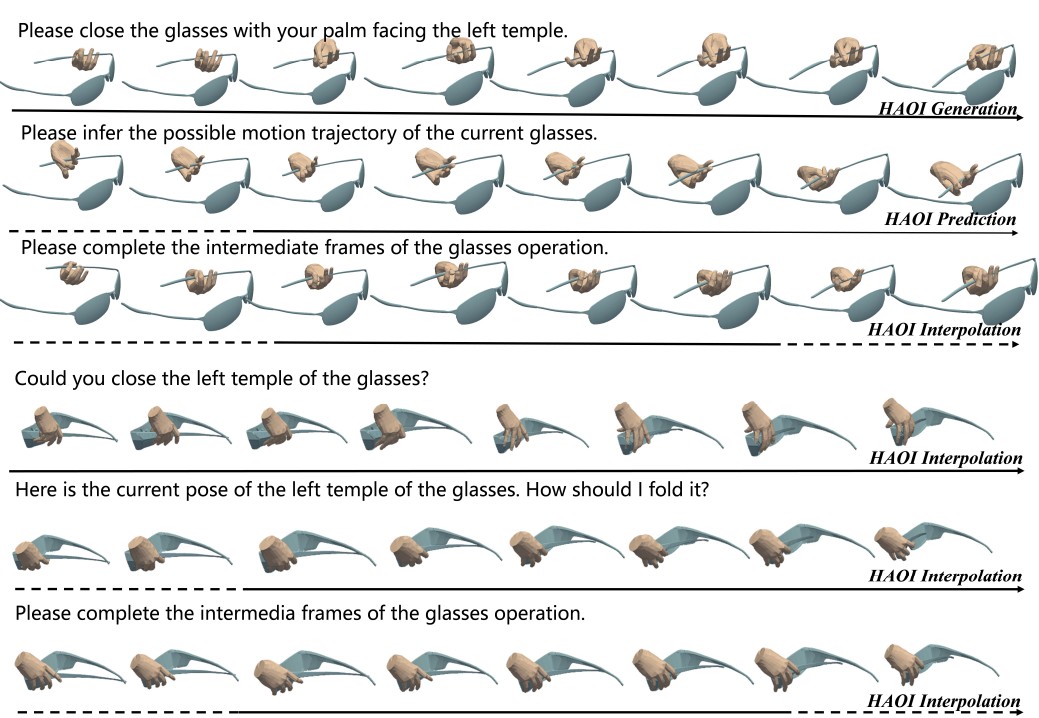

Figure 9: HAOI generation, prediction, and completion on small-to-medium objects: glasses.

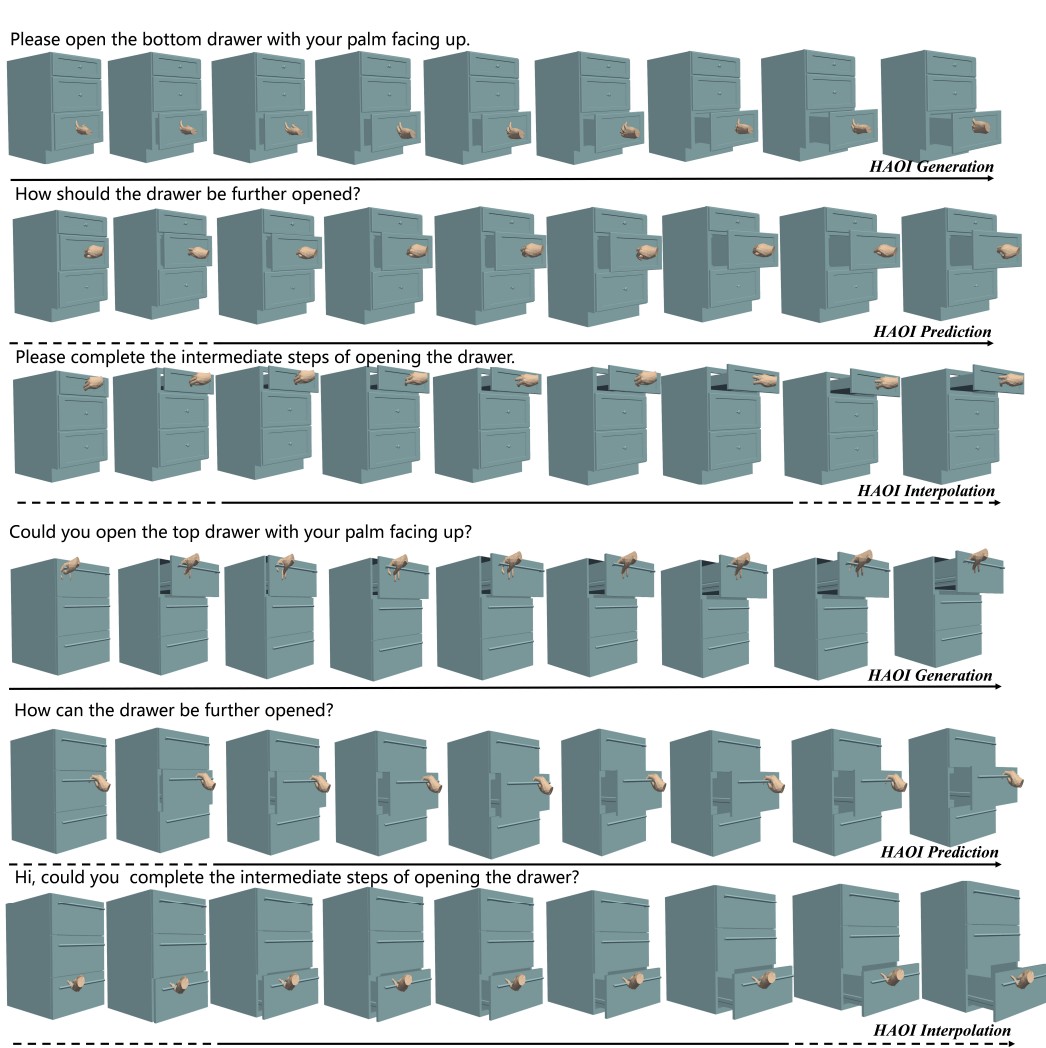

Figure 10: HAOI generation, prediction, and completion on large objects: cabinet.

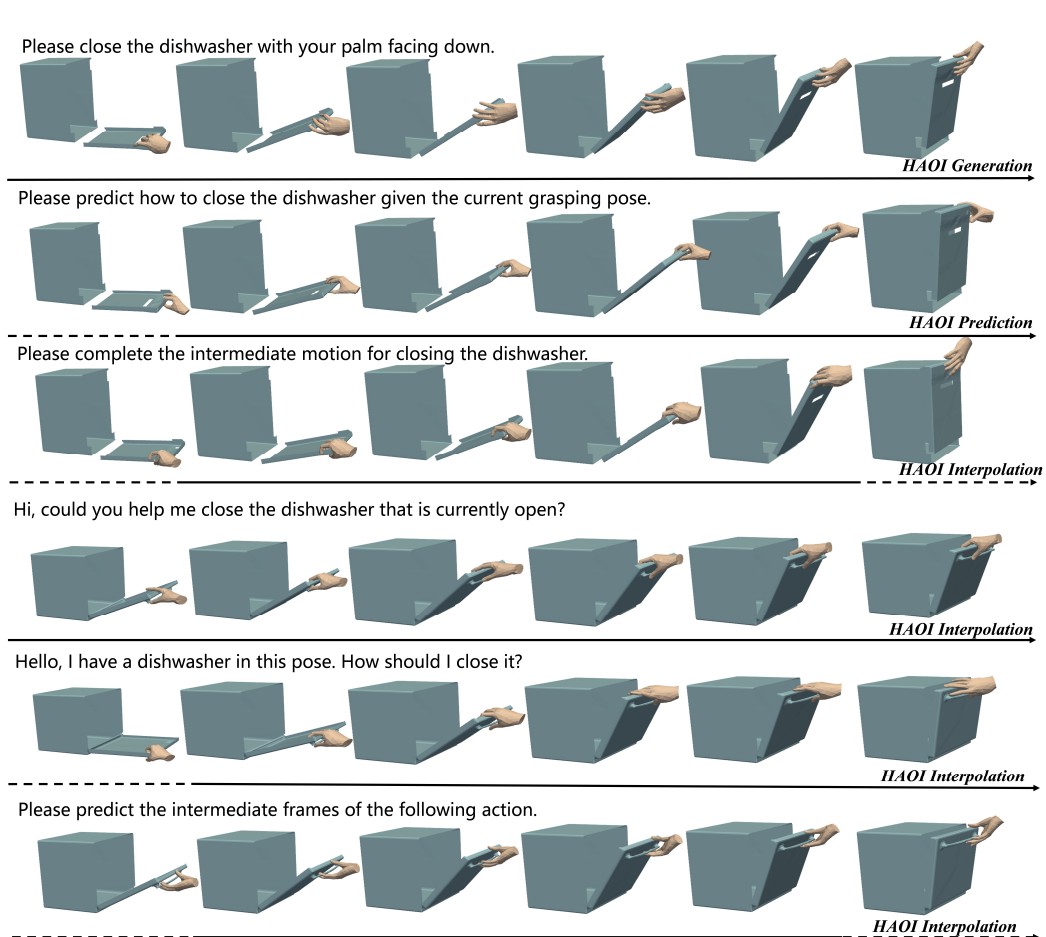

Figure 11: HAOI generation, prediction, and completion on large objects: dishwasher.

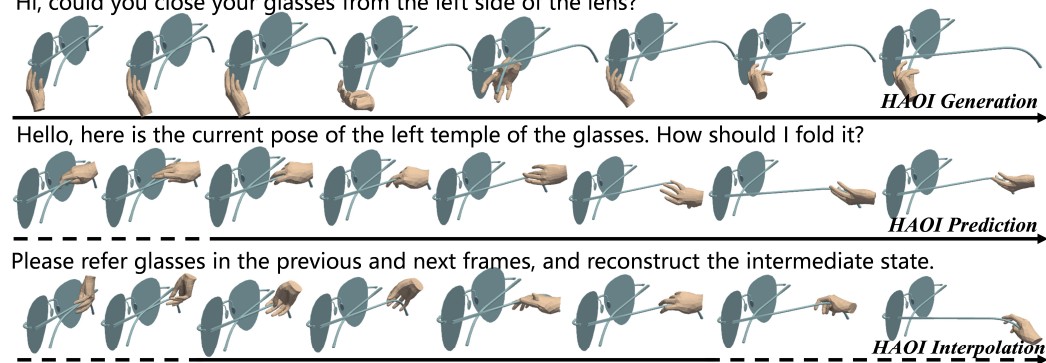

Figure 12: Representative failure case of our model.

