# OpenReview forum: "SynHLMA:Synthesizing Hand Language Manipulation for Articulated Object with Discrete Human Object Interaction Representation"
_ICLR.cc/2026/Conference — Submitted to ICLR 2026_

### Official Review · Reviewer_ENDg · 2025-10-30

**Soundness:** 3
**Presentation:** 1
**Contribution:** 3
**Rating:** 4
**Confidence:** 4

**Summary:**

This work presents SynHLMA, a novel framework for synthesizing Hand Articulated Object Interaction (HAOI) sequences from natural language instructions. A key contribution is the HAOI-Lang dataset, a new, large-scale benchmark of physically plausible manipulation sequences generated via reinforcement learning and paired with rich language annotations. The proposed method models the problem by first leveraging a multi-stage VQ-VAE to discretize the continuous, long-term HAOI trajectories into hierarchical semantic tokens. These discrete representations are then aligned with language embeddings in a shared semantic space using an HAOI Manipulation Language Model. A novel articulation-aware loss is introduced to ensure the generated hand motions are consistent with the object's joint dynamics. Experimental results demonstrate that SynHLMA achieves state-of-the-art performance on HAOI generation, prediction, and interpolation tasks.

**Strengths:**

This paper has the following strengths:

A significant contribution is the construction of the HAOI-Lang dataset, a new benchmark for Human Articulated Object Interactions. This dataset was generated using reinforcement learning in a physics simulator and is comprehensive, featuring 256 instances with over 500,000 static grasps and 50,000 manipulation sequences. This is helpful for the community.

The paper successfully applies a multi-stage VQ-VAE framework to the HAOI task. This approach effectively discretizes the continuous interactions, enabling the generation of corresponding HAOI motion sequences directly from human language instructions.

The proposed method, SynHLMA, demonstrates state-of-the-art (SOTA) performance on the challenging HAOI generation task, outperforming existing baselines.

The strategy of tokenizing the interaction data and processing it with a Large Language Model (LLM) is effective. It successfully fuses natural language semantics with the LLM's priors and is shown to be beneficial for generating diverse outputs.

**Weaknesses:**

The paper has several weaknesses that should be addressed:

Presentation Quality: The overall presentation quality is lacking. There are visual issues, such as formatting (e.g., the title of Section 4.3) and poor figure design. For instance, Figure 2 is overly crowded, with elements like the 'Co' label being obscured, despite large amounts of unused white space on the figure's flanks. Furthermore, the writing could be improved; Section 3.3, for example, lacks a clear narrative structure, making the methodology difficult to follow.

Limited Novelty: The paper's core methodology, which relies on a multi-stage VQ-VAE, appears to have limited novelty. This framework has been established in prior work. The current paper seems to be a direct application of this existing framework to the HAOI generation task, with insufficient task-specific innovation.

Insufficient Qualitative Results: The evaluation of the generated results is not comprehensive. Without a video supplement, it is difficult to intuitively assess the quality, fluency, and physical plausibility of the generated motions. Moreover, while the dataset contains 256 instances, the paper and appendix only provide detailed and dense sequence visualizations for a very small subset (only 4 objects in the appendix). This limited sample size is insufficient to fully demonstrate the model's true performance and generalization capabilities.

**Questions:**

Generalization: How well does the model generalize to new scenarios? Specifically, can it handle unseen object instances or entirely new object categories? Can it perform novel manipulation tasks (e.g., actions not present in the training data)? Furthermore, how robust is its generalization to language instructions that specify novel contact locations?

Motion Consistency and Continuity: Do the generated manipulation sequences maintain good temporal consistency and continuity? For example, do the contact points drift unreasonably during the operation? Does the hand pose (gesture) change illogically, or are the generated trajectories unnatural or jittery? It would be beneficial if the authors could provide and analyze some typical failure cases.

Physical Plausibility and Real-World Transfer: Are the generated motions physically plausible? The paper shows a simulation transfer, but can these generated sequences be successfully and directly applied to a physical, real-world robotic hand? What would be the main challenges?

---

> ### Author Response · Authors · 2025-11-18
> **Rebuttal by Authors**
>
> We sincerely appreciate your valuable comments. Below we provide a point-by-point response to your questions. If you have any additional concerns, please feel free to let us know. We genuinely look forward to further discussion with you.
>
> ### 1. Writing Issues
>
> We sincerely appreciate your insightful comments. We have carefully re-read the entire paper and made substantial revisions. Specifically:
>
> In Section 3.3, we have reorganized the presentation flow by first introducing the VQ-VAE responsible for joint prediction, followed by the multi-level VQ-VAE used for reconstructing HAOI grasping parameters. We then separately describe the first component of the articulation-aware loss and the loss functions used in the multi-level VQ-VAE. In Section 3.4, we provide a more detailed explanation of the articulation-aware loss.
>
> We have also refined Fig. 2 to address the occlusion issue related to the tensor $C_\mathcal{O}$. In addition, we include several new ablation studies (e.g., articulation-aware loss, comparisons with Qwen and Gemma, continuous vs. discrete feature representations), which together make the paper more comprehensive and convincing.
>
> We add an analysis of failure cases in the appendix to further enhance the completeness of the work.
>
> Finally, We corrected formatting errors throughout the manuscript.
>
> ---
> ### 2. Concerns Regarding Novelty
>
> Thank you very much for your professional feedback. Compared with traditional multi-stage VQ-VAE generation, our method introduces a Joint Predict Decoder to explicitly model joint states, enabling higher-quality and finer-grained static HAOI generation.
>
> Furthermore, to ensure physical plausibility in VQ-VAE HAOI reconstruction and consistency/continuity in the manipulation sequence produced by the Manipulation Language Model (MLM), we propose an **articulation-aware** loss tailored for articulated objects. Standard VQ-VAE relies solely on reconstruction loss and cannot guarantee consistency between grasp and joint dynamics.
>
> Our loss design explicitly encodes both static HAOI structure and continuous manipulation motion:
> - In **Stage 1**, for per-frame static HAOI reconstruction, we employ **HAOI Penetration Loss** and **Joint-Aware Loss** to ensure physical plausibility and correct joint-state awareness.
> - In **Stage 2**, after the MLM outputs tokens for each frame, the multi-stage VQ-VAE decodes hand and object rotation matrices. We compute inter-frame changes and enforce pose consistency using the temporal constraint term.
>
> To the best of our knowledge, this is the first work to introduce an articulation-aware loss into a discrete VQ-VAE framework for HAOI generation, explicitly enforcing joint-state consistency across both spatial and temporal dimensions.
>
> We will sequentially present the results: first, a comparison with SemGrasp (traditional multi-stage VQ-VAE generation), followed by an ablation study on the articulation-aware loss.
>
> #### **Comparison with SemGrasp**
> | Methods | FID ↓ | Diversity → | MMDist ↓ | IV ↓ | ADE ↓ | FDE ↓ |
> |--------|-------|-------------|----------|------|-------|-------|
> | Real | 0.002 | 43.427 | - | - | - | - |
> | SemGrasp | 20.873 | 27.954 | 16.482 | 10.247 | 1.284 | 1.932 |
> | **SynHLMA (Ours)** | **14.121** | **40.484** | **12.793** | **5.919** | **0.976** | **1.147** |
>
> These results demonstrate that the articulation-aware loss effectively links key components of the model and explicitly encodes both static HAOI states and continuous manipulation dynamics, enabling high-quality articulated-object sequence generation.
>
> #### **Ablation on Articulation-Aware Loss**
>
> | Setting | FID ↓ | IV ↓ | ADE ↓ |
> |---------|--------|---------|----------|
> | w/o $𝓛_{artic}$ | 15.872 | 6.452 | 1.041 |
> | w/o $𝓛_C$ | 14.990 | 6.104 | 1.112 |
> | w/o $𝓛_{artic}$ ∧ $𝓛_C$ | 16.840 | 6.987 | 1.233 |
>
> In the above table, we provide a detailed evaluation of the proposed articulation-aware loss designed for articulated objects. The experimental results show that:
>
> 1. Without the static-frame constraint term $𝓛_{artic}$, both IV and FID increase significantly, indicating reduced physical plausibility and contact consistency in HAOI grasping, as well as larger errors in the generated grasp poses.
> 2. When removing $𝓛_C$, the ADE rises markedly, suggesting that temporal inconsistencies—such as noticeable "jittering" or "discontinuities" in hand-object angular transitions—become more pronounced.
> 3. When both loss terms are ablated together, the overall sequence generation quality drops to the worst level.
>
> These results demonstrate that the proposed articulation-aware loss effectively links the key components of the model and explicitly encodes both static HAOI states and continuous manipulation dynamics, thereby ensuring high-quality generation.
>
>
> ---

---

> ### Author Response · Authors · 2025-11-18
> **Rebuttal by Authors**
>
> ### 3. Qualitative Results and Motion Consistency/Continuity
>
> We appreciate your insightful suggestions. In Appendix A.5, we added visualizations for four additional instances. Due to space limitations, we cannot include more examples within the main paper, but after acceptance, we plan to release an interactive visualization tool or website for large-scale qualitative inspection. More importantly, for the sake of visualization, we generated **demo videos** of the laptop and cabinet being opened and included them in the supplementary materials.
>
> In Appendix A.6, we provide representative failure cases:
>
> - **HAOI generation**: When the manipulation description is overly strict or physically implausible, the generated grasp may deviate from realistic configurations.
> - **HAOI prediction**: The predicted interaction sequence may suffer from temporal inconsistencies, such as hand drift or deformation over time. For example, between frames 3 and 8, although the average penetration error is 0.391 cm, the final rotation error reaches 0.748 rad.
> - **HAOI interpolation**: Abrupt hand-pose transitions may appear due to discontinuities in the interpolated pose sequence. Within the interval from frame 3 to frame 8, the penetration error peaks at 0.467 cm, while the average rotation error is 0.241 rad.
>
> ---
>
> ### 4. Model Generalization to New Scenarios
>
> Thank you for your insightful question. Our model is category-aware, meaning it generalizes to different object instances within the same category.
>
> For novel *categories*, our first idea is few-shot learning. However, extending VQ-VAE with few-shot learning typically requires freezing both the encoder and the codebook. Because our model uses category-aware tokens—where each object’s discrete features are required during training and inference—few-shot learning alone is insufficient when intra-category shape variations are large.
>
> Inspired by PA-Diffusion, we plan to introduce a **primitive prototype library**, which may facilitate part-level cross-category generalization in future work.
>
> Regarding *novel grasping modes* or *entirely new grasp points*:
> VQ-VAE’s discrete codebook provides robustness and reusable motion patterns but generalizes poorly to unseen grasp strategies.
> Reasons include:
> 1) Novel grasp modes may be quantized into incorrect codebook entries;
> 2) VQ-VAE is inherently unsuitable for out-of-distribution extrapolation.
>
> ---
>
> ### 5. Transfer to the Real World
>
> We sincerely appreciate your forward-looking question. Dexterous grasping in simulation is already well-established, and many recent works have demonstrated successful sim-to-real transfer.
>
> Traditional domain adaptation often requires abundant unlabeled real-world data, while domain randomization may waste simulation fidelity. Recent advances reduce data requirements significantly. For example, **Sim-to-Real via Sim-to-Sim** unifies simulated and real inputs into a canonical space for grasping, reducing real-world data usage by >99% while achieving **91% real-world grasp success** using only **5000** real samples.
>
> For dexterous manipulation, **DexPoint** proposes a generalizable point-cloud reinforcement learning approach by combining observed point clouds, imagined point clouds, robot proprioception, and target poses into a unified representation.
>
> Our method can guide dexterous grasping in simulation and can be integrated into existing sim-to-real pipelines.
> In our HAOI-lang simulation, the grasp accuracy is approximately:
> - **Midpoint Error**: ~3.96 cm
> - **Wrist Rotation Error**: ~0.451 rad
>
> These results show that our model's predicted grasp sequences are accurate and physically plausible, providing a strong foundation for sim-to-real transfer.

---

> ### Author Response · Authors · 2025-11-24
>
> Dear reviewer, we sincerely appreciate your thoughtful and valuable comments. We have provided a point-by-point response, and we truly hope that our clarifications address your concerns. Please let us know if there are any remaining issues—we are fully committed to resolving them before the discussion period ends. Your feedback means a great deal to us.

---

> ### Author Response · Authors · 2025-11-28
>
> Dear Reviewer, we sincerely appreciate the time and care you have devoted to evaluating our submission. With the discussion period concluding in a few days, we would like to gently inquire whether you have any remaining questions or if there are further analyses or experiments you would like us to explore. We are more than willing to provide additional theoretical insights or experimental evidence promptly to help clarify any aspects that may still be unclear. Your feedback is extremely important to the development of our work, SynHLMA, and we would be deeply grateful for your continued consideration.

---

### Official Review · Reviewer_nYnu · 2025-11-01

**Soundness:** 3
**Presentation:** 3
**Contribution:** 3
**Rating:** 6
**Confidence:** 3

**Summary:**

The paper introduces SynHLMA, a new framework for synthesizing hand-articulated object manipulation sequences from natural language instructions. The method discretizes each frame of a hand-object interaction into hierarchical VQ-VAE tokens, encoding global hand pose, local articulation, refinement, and joint state, and trains a manipulation language model (Vicuna-7B with LoRA) to generate, predict, and interpolate Human Articulated Object Interactions (HAOI). A new dataset, HAOI-Lang, is built using physics simulation (RaiSim) and GPT-4 language annotations, comprising 50k manipulation sequences over seven object categories. Experiments show clear gains over HOIGPT and Text2HOI in FID, diversity, and ADE across three tasks (generation, prediction, interpolation), and qualitative transfer to the ShadowHand robot demonstrates physical plausibility.

**Strengths:**

1. Solid methodology integrating VQ-VAE and LoRA-tuned Vicuna. Consistent improvements on benchmarks.

2. Dataset Contribution: HAOI-Lang offers valuable large-scale multimodal data with physics-consistent interactions and generated instructions.

3. Generality: Demonstrated applicability across generation, prediction, and interpolation tasks. Showed extension to robotic dexterity transfer.

4. Clarity: Good pipeline and ablation studies on design choices (token hierarchy, VQ-VAE size, LoRA rank).

**Weaknesses:**

1. The dataset is simulation-based, and language annotations are GPT-generated, which may limit transfer to real-world hand motions or linguistic diversity.

2. Robotic transfer is only qualitative within a simulator; no human demonstration or physical validation.

3. Comparative Scope: Comparisons are mainly against HOIGPT/Text2HOI; no baselines using diffusion or transformer-based generative models (e.g., AffordanceDiffusion, HOIDiffusion, NL2Contact).

4. Ablation Breadth: Although token-level ablations are detailed, no explicit study isolates the effect of articulation-aware loss or discrete vs. continuous representations.

5. Dataset Documentation: bias considerations of using GPT-4-generated text (e.g., annotation consistency, cultural phrasing) are not discussed.

**Questions:**

1. How sensitive is performance to the choice of Vicuna-7B vs. other LLMs (e.g., Qwen2 or LLaVA)?

2. Can the articulation-aware loss generalize to multi-joint or compound-joint objects beyond single revolute/prismatic types?

3. How does SynHLMA perform when applied to real RGB-D or point-cloud sequences captured from human manipulation instead of simulation?

4. Could discrete HAOI tokens be shared across object categories to enable few-shot generalization?

5. What are the failure modes in complex or bimanual manipulations (e.g., eyeglass folding, cabinet doors with two handles)?

6. Are there plans to release the dataset with verified physical realism or human-validated captions?

---

> ### Author Response · Authors · 2025-11-18
> **Rebuttal by Authors**
>
> We sincerely appreciate your valuable comments. Below we provide a point-by-point response to your questions. If you have any additional concerns, please feel free to let us know. We genuinely look forward to further discussion with you.
>
> ### 1. On the Sim-to-Real Transfer Problem
>
> We sincerely appreciate your question. Simulating dexterous-hand grasping in physics engines is already a well-established research area. In recent years, many works have also explored sim-to-real transfer and achieved promising results. Sim-to-real must overcome challenges such as real-world occlusions and unknown object physical properties. In fact, the core difficulty lies in the *lack of real-world grasp data*, even though grasping is one of the most common human actions.
>
> Traditional domain adaptation requires a large amount of unlabeled real-world data, and domain randomization alone may underutilize structured virtual data. Some recent approaches instead standardize the scene—for example, *Sim-to-Real via Sim-to-Sim* converts both simulator and real camera observations into a unified canonical space. This dramatically reduces data demand: with only **5,000 real grasps** for joint finetuning, the method achieves **91% real-world grasp success**, comparable to a system trained on **580,000** real grasps, reducing real data usage by over **99%**.
>
> For dexterous hands, *DexPoint* proposes a generalizable point-cloud RL approach. Its feature extractor takes as input observed point clouds, imagined point clouds, robot self-perception, and target poses, and outputs a unified embedding. Reinforcement learning then guides the policy to acquire real-world grasping skills.
>
> Given the maturity of these sim-to-real techniques, our method—already capable of generating high-quality grasps in simulation—can also be transferred to real robot hands through similar mechanisms. In the HAOI-lang simulation experiments, our model achieves a **Midpoint Error of ~3.96 cm** and a **Wrist Rotation Error of ~0.451 rad**.
>
> ---
>
> ### 2. On the Comparison Experiments
>
> Thank you for your thoughtful question. In Sec. 4.2, we added comparisons against **NL2Contact**. NL2Contact uses conditional diffusion for coarse grasp synthesis followed by an optimization-based refinement. In contrast:
>
> - We use the **Joint Prediction Decoder** to explicitly model joint states.
> - We adopt a **multi-stage VQ-VAE** to generate HAOI across three stages (global → local → refinement).
> - Through the **articulation-aware loss**, we impose explicit constraints on token modeling and temporal output consistency.
>
> To the best of our knowledge, this is the first work to introduce an articulation-aware loss into a discrete VQ-VAE framework for HAOI generation, explicitly enforcing joint-state consistency across both spatial and temporal dimensions. (Please refer to your next question for detailed loss analysis.)
> Below are the comparison results:
>
> #### **Comparison with NL2Contact**
>
> | Methods | FID ↓ | Diversity → | MMDist ↓ | IV ↓ | ADE ↓ | FDE ↓ |
> |--------|--------|--------------|-----------|--------|-----------|-----------|
> | Real | 0.002 | 43.427 | - | - | - | - |
> | NL2Contact | 25.610 | 19.843 | 19.127 | 12.984 | 1.742 | 3.865 |
> | **SynHLMA** | **14.121** | **40.484** | **12.793** | **5.919** | **0.976** | **1.147** |
>
> ---

---

> ### Author Response · Authors · 2025-11-18
> **Rebuttal by Authors**
>
> ### 3. On the Ablation Study
>
> Thank you for the valuable comments. To ensure both physical plausibility in VQ-VAE-based HAOI generation and temporal consistency in manipulation-sequence decoding, we introduce a set of **articulation-aware losses**. A vanilla VQ-VAE with only reconstruction loss cannot enforce grasp–joint consistency or articulation dynamics.
>
> Our loss design explicitly encodes both static HAOI structure and continuous manipulation motion:
>
> - **Stage 1 (per-frame generation):**
>   We apply **HAOI Penetration Loss** and **Joint Aware Loss** to enforce physical plausibility and semantic correctness of the articulated joints.
>
> - **Stage 2 (sequence generation):**
>   After the manipulation language model predicts tokens for each frame, the multi-stage VQ-VAE decodes hand–object rotation matrices. We compute per-frame transitions and enforce **pose consistency** by matching adjacent-frame changes.
>
> Ablation results for articulation-aware loss are shown below:
>
> #### **Ablation on Articulation-Aware Loss**
>
> | Setting | FID ↓ | IV ↓ | ADE ↓ |
> |---------|--------|---------|----------|
> | w/o $𝓛_{artic}$ | 15.872 | 6.452 | 1.041 |
> | w/o $𝓛_C$ | 14.990 | 6.104 | 1.112 |
> | w/o $𝓛_{artic}$ ∧ $𝓛_C$ | 16.840 | 6.987 | 1.233 |
>
> In the above table, we provide a detailed evaluation of the proposed articulation-aware loss designed for articulated objects. The experimental results show that:
>
> 1. Without the static-frame constraint term $𝓛_{artic}$, both IV and FID increase significantly, indicating reduced physical plausibility and contact consistency in HAOI grasping, as well as larger errors in the generated grasp poses.
> 2. When removing $𝓛_C$, the ADE rises markedly, suggesting that temporal inconsistencies—such as noticeable "jittering" or "discontinuities" in hand-object angular transitions—become more pronounced.
> 3. When both loss terms are ablated together, the overall sequence generation quality drops to the worst level.
>
> These results demonstrate that the proposed articulation-aware loss effectively links the key components of the model and explicitly encodes both static HAOI states and continuous manipulation dynamics, thereby ensuring high-quality generation.
>
> In Sec. 4.3, to verify the advantage of the **discrete multi-stage VQ-VAE representation** over continuous AE representations, we replace all VQVAE modules with AEs while keeping network size and multi-stage structure unchanged. Results show that discrete codebooks naturally cluster motion primitives (grasping, rotation, translation), providing stable and structured representations. VQ-VAE avoids mode collapse and supports high-complexity sequence generation.
>
> #### **VQ-VAE vs. AE (Discrete vs. Continuous Representation)**
>
> | Setting | FID ↓ | Diversity ↑ | ADE ↓ |
> |---------|--------|--------------|----------|
> | w/o ⟨g,l,r,j⟩ (AE) | 1.160 | 3.411 | 1.012 |
> | **⟨g,l,r,j⟩ (Ours)** | **0.699** | 3.109 | **0.815** |
>
> ---
> ### 4. On the Use of GPT
>
> Thank you for your insightful perspective. After evaluating captioning capabilities of multiple MLLMs, we ultimately chose ChatGPT. Thanks to its strong visual reasoning and motion-analysis ability, ChatGPT generates high-quality captions even when the video resolution is reduced for speed and storage reasons.
>
> Below we include a direct comparison between GPT and Gemma. GPT clearly produces more realistic, detailed, and interaction-aware descriptions:
>
> **ChatGPT-4:**
> *The action shows the hand approaching the laptop lid from the upper-right, making a light edge contact, then forming a pinch grip to lift it. As the lid opens, the hand slides toward the right corner, the wrist rotates to match the lid’s angle, and the fingers shift onto the top surface for stability. The motion ends with a light guiding hold as the lid reaches its fully open position.*
>
> **Gemma:**
> *This video shows a person holding a laptop screen and gradually opening it from a closed state. In the image on the far left, the screen is almost completely closed, and in the following images, it gradually rotates upwards until, in the image on the far right, the screen is fully open and at a usable angle.*
>
> ---
>
> ### 5. Ablation on Different MLLM Backbones
>
> Thank you for the professional suggestion. In Sec. 4.3, we added comparisons with **Qwen2.5-7B-Instruct** and **Gemma-7B-it**.
>
> - **Qwen** yields weaker results, possibly due to weaker fine-grained motion reasoning or due to our dataset captions being entirely in English.
> - **Gemma** performs similarly to Vicuna, but since Gemma’s base model is larger, we were only able to train it using **4-bit quantization**, whereas Vicuna was trained in **8-bit**, resulting in a slight performance gap.
>
> #### **Ablation on Manipulation Language Model Backbone**
>
> | Backbone | FID ↓ | MMDist ↓ | IV ↓ |
> |-----------|----------|--------------|-----------|
> | w/ LLaMA | 51.911 | 17.222 | 9.244 |
> | w/ Qwen | 73.590 | 25.272 | 6.848 |
> | w/ Gemma | 22.576 | 16.228 | 2.317 |
>
> ---

---

> ### Author Response · Authors · 2025-11-18
> **Rebuttal by Authors**
>
> ### 6. On the Articulation Loss Design
>
> Thank you for this insightful question. Unfortunately, the current articulation-aware loss can only be applied to **single-object articulation**. The key limitation lies in the **pose consistency loss $\mathcal{L}_C$**. Due to the fundamentally different motion patterns of rotational and translational joints, there is no unified formulation that handles both reliably.
>
> For special cases such as bottle caps, rotation naturally induces translation (e.g., upward displacement when twisting open). In such cases, translation-based consistency may be applicable. More general formulations capable of handling arbitrary joint types remain an open research question.
>
> ---
>
> ### 7. On Real-World RGB-D and Point Clouds Data
> We sincerely thank the reviewer for raising this important question. We would like to clarify that the ArtImage dataset used in our work contains **real scanned objects**, so the object geometry is already consistent with real-world shapes. Although the manipulation trajectories in HAOI-Lang are sampled in simulation, the objects themselves originate from real captures.
>
> Regarding evaluation on real RGB-D or point-cloud human manipulation sequences, while several real-world HOI datasets exist (e.g., GRAB, ARCTIC), they are unfortunately **not directly applicable** for quantitative comparison in our setting for the following reasons:
>
> 1. **Limited overlap in object categories.**
>    As discussed in our response to Question 8, cross-category evaluations introduce significant performance drop. GRAB and ARCTIC share very few object categories with ours, making direct evaluation uninformative regarding robustness.
>
> 2. **Fundamental differences in manipulation distributions.**
>    Even when occasional object categories coincide, the manipulation goals in GRAB/ARCTIC and in our framework may align, yet the executed actions are different. The manipulation goals, grasping behaviors, and action distributions differ substantially. Therefore, making one-to-one comparison neither suitable nor meaningful.
>
> We would like to emphasize that our experiments already demonstrate strong generalization across different object instances within the same category, indicating good intra-class robustness.
>
> We also plan to sample real-world data for HAOI manipulation generation in future work. Depending on the type of data, the following approaches may be considered:
>
> - **For RGB-D input:** A lightweight visual analysis module would be required for object-feature extraction, while the rest of our pipeline can remain unchanged. A potential challenge is insufficient perception of occluded regions or object scale.
>
> - **For real-world point-cloud input:** Our framework can be applied almost directly with minimal or no modification.
>
> ---
>
> ### 8. On Token Generalization
>
> Thank you for your valuable perspective.
> Our model is category-aware, meaning it generalizes to different object instances within the same category.
>
> For novel *categories*, our first idea is few-shot learning. However, extending VQ-VAE with few-shot learning typically requires freezing both the encoder and the codebook. Because our model uses category-aware tokens—where each object’s discrete features are required during training and inference—few-shot learning alone is insufficient when intra-category shape variations are large.
>
> Inspired by PA-Diffusion, we plan to introduce a **primitive prototype library**, which may facilitate part-level cross-category generalization in future work.
>
> Regarding *novel grasping modes* or *entirely new grasp points*:
> VQ-VAE’s discrete codebook provides robustness and reusable motion patterns but generalizes poorly to unseen grasp strategies.
> Reasons include:
> 1) Novel grasp modes may be quantized into incorrect codebook entries;
> 2) VQ-VAE is inherently unsuitable for out-of-distribution extrapolation.
>
> ---
> ### 9. On Failure Modes of Two-Hand Generation
>
>  Thank you for your constructive feedback. The generation quality of our method primarily depends on two key components of the network: the VQ-VAE and the MLLM. In particular, for the MLLM, its performance tends to degrade when it is applied directly to tasks for which no prior knowledge exists and which are not covered during fine-tuning. We attempted to evaluate it on complex bi-manual manipulation tasks, but the manipulation language model failed to produce outputs that conformed to our required format. We plan to extend our work to two-hand generation. Our current model can also be adapted: for example, if generating right-hand motion, the left hand can share the same multi-stage VQ-VAE architecture. If time permits, we will include a demo showing qualitative results.
>
> ---
> ### 10. On Dataset Release
>
> We sincerely appreciate your interest. Our plan is to release:
>
> - the simulator scripts used for generating HAOI,
> - the visualization tools,
> - and the full dataset.
>
> Please stay tuned.

---

> ### Author Response · Authors · 2025-11-24
>
> Dear reviewer, we sincerely appreciate your thoughtful and valuable comments. We have provided a point-by-point response, and we truly hope that our clarifications address your concerns. Please let us know if there are any remaining issues—we are fully committed to resolving them before the discussion period ends. Your feedback means a great deal to us.

---

> ### Author Response · Authors · 2025-11-28
>
> Dear Reviewer, we sincerely appreciate the time and care you have devoted to evaluating our submission. With the discussion period concluding in a few days, we would like to gently inquire whether you have any remaining questions or if there are further analyses or experiments you would like us to explore. We are more than willing to provide additional theoretical insights or experimental evidence promptly to help clarify any aspects that may still be unclear. Your feedback is extremely important to the development of our work, SynHLMA, and we would be deeply grateful for your continued consideration.

---

### Official Review · Reviewer_9XZF · 2025-11-01

**Soundness:** 3
**Presentation:** 3
**Contribution:** 3
**Rating:** 6
**Confidence:** 4

**Summary:**

The paper presents SynHLMA, a language-conditioned framework for human–articulated-object interaction. Each frame is discretized into hierarchical tokens via a multi-stage VQ-VAE, and a joint-aware loss enforces articulation consistency. A manipulation language model aligns the tokenized sequences with natural-language instructions, enabling long-horizon generation, prediction, and interpolation of hand–object trajectories. The authors also release HAOI-Lang, a large simulated dataset with GPT-annotated language, and report consistent improvements over strong HOI/motion baselines on FID, diversity, ADE/FDE, and related metrics, complemented by qualitative results. Finally, they demonstrate transfer to dexterous robotics by fitting synthesized MANO trajectories to a ShadowHand for imitation.

**Strengths:**

1. The dynamic modeling of hand interactions with articulated objects is well motivated and encodes both semantic intent and articulation constraints in a coherent formulation.
2. The curated dataset is likely to be useful for the community and may enable controlled studies of language-conditioned articulated manipulation.
3. The experimental evaluation in simulation is reasonably comprehensive, covering generation, prediction, and interpolation with quantitative and qualitative evidence.

**Weaknesses:**

1. Rendering-to-GPT caption pipeline. The paper relies on rendering sequences in Open3D and obtaining descriptions with GPT-4. The realism of Open3D renderings is limited, which may introduce a domain gap for image-to-text captioning and, in turn, for language supervision quality.
2. Assumption on fixed object base. It is unclear whether the method supports scenarios in which the articulated object’s base moves in the world. The token index \<j\> is defined in the object’s canonical space, which may implicitly assume a fixed base (e.g., a laptop fixed on a table), potentially excluding sequences like “pick up the laptop, then close the lid.”
3. Comparative analysis with SemGrasp. Although SemGrasp targets single-step semantic grasping rather than long-horizon manipulation, a comparison or discussion would help position the contribution relative to semantic grasp baselines.

**Questions:**

The following questions are based on the weaknesses discussed above; please refer to that section.

---

> ### Author Response · Authors · 2025-11-18
> **Rebuttal by Authors**
>
> We sincerely appreciate your valuable comments. Below we provide a point-by-point response to your questions. If you have any additional concerns, please feel free to let us know. We genuinely look forward to further discussion with you.
>
> ### 1. Rendering Issues of Open3D
>
> Thank you for your insightful comments. Indeed, the rendering quality of Open3D is sometimes not fully realistic, and for reasons of processing speed and disk storage, we further compress the output video, which may impact its visual fidelity. However, benefitting from ChatGPT’s extensive prior knowledge, the captions generated by ChatGPT can accurately reflect the interaction between the hand and articulated object throughout the HAOI motion sequence.
>
> To better illustrate why we choose ChatGPT for caption generation, we include below several descriptions from other MLLMs for the same video. As shown, ChatGPT-4 provides descriptions that are more natural and comprehensive.
>
> **ChatGPT-4:**
> *The action shows the hand approaching the laptop lid from the upper-right, making a light edge contact, then forming a pinch grip to lift it. As the lid opens, the hand slides toward the right corner, the wrist rotates to match the lid’s angle, and the fingers shift onto the top surface for stability. The motion ends with a light guiding hold as the lid reaches its fully open position.*
>
> **Gemma:**
> *This video shows a person holding a laptop screen and gradually opening it from a closed state. In the image on the far left, the screen is almost completely closed, and in the following images, it gradually rotates upwards until, in the image on the far right, the screen is fully open and at a usable angle.*
>
> ---
>
> ### 2. Assumption Regarding the Fixed Object Base
>
> We sincerely appreciate your valuable feedback. As you correctly pointed out, our token index `<j>` is defined under the assumption that the object is placed in a canonical space. We do **not** assign `<j>` any semantics related to global object translation; instead, it is used purely to model the joint angle of the articulated part.
>
> The action you mentioned — *“pick up the laptop, then close the lid”* — is more representative of **two-hand** interaction. We indeed plan to extend our framework to two-hand manipulation generation in future work, possibly based on datasets such as ARCTIC or by collecting two-hand interaction data. At that time, the model will be modified accordingly to accommodate richer interaction semantics in HAOI.
>
> ---
>
> ### 3. Comparison with SemGrasp
>
> We sincerely thank you for your constructive suggestions. SemGrasp focuses on generating **static** HOI. Building upon this, our method further proposes the generation of **HAOI manipulation sequences**.
>
> Compared with SemGrasp, our approach models joint states explicitly through the **Joint Prediction Decoder**, enabling higher-quality and more fine-grained static HAOI generation through the multi-stage VQ-VAE. Moreover, to ensure both physical plausibility in VQ-VAE reconstruction and temporal consistency in the manipulation language model, we introduce a series of **articulation-aware losses**. A standard VQ-VAE relying solely on reconstruction loss cannot guarantee grasp–joint consistency or accurate articulation dynamics.
>
> Specifically:
>
> - **Stage 1 (per-frame modeling):**
>   We apply **HAOI Penetration Loss** and **Joint Aware Loss** to ensure physical plausibility and correct articulation perception in each static HAOI frame.
>
> - **Stage 2 (manipulation sequence modeling):**
>   After the manipulation language model outputs the token sequence for each frame, the multi-stage VQ-VAE decodes hand and object rotation matrices. We compute the transition values between adjacent frames and enforce **temporal consistency** between hand pose and object articulation.
>
> To the best of our knowledge, this is the first work to introduce an articulation-aware loss
> into a discrete VQ-VAE framework for HAOI generation, explicitly enforcing joint-state
> consistency across both spatial and temporal dimensions. In Sec. 4.2 of the main paper, we included a detailed comparison with SemGrasp. The results are shown below:
>
> #### **Comparison with SemGrasp**
>
> | Methods | FID ↓ | Diversity → | MMDist ↓ | IV ↓ | ADE ↓ | FDE ↓ |
> |---------|--------|--------------|------------|---------|----------|----------|
> | Real | 0.002 | 43.427 | - | - | - | - |
> | SemGrasp | 20.873 | 27.954 | 16.482 | 10.247 | 1.284 | 1.932 |
> | **SynHLMA** | **14.121** | **40.484** | **12.793** | **5.919** | **0.976** | **1.147** |
> ---
>
> These results demonstrate that the articulation-aware loss effectively links key components of the model and explicitly encodes both static HAOI states and continuous manipulation dynamics, enabling high-quality articulated-object sequence generation.

---

> ### Author Response · Authors · 2025-11-24
>
> Dear reviewer, we sincerely appreciate your thoughtful and valuable comments. We have provided a point-by-point response, and we truly hope that our clarifications address your concerns. Please let us know if there are any remaining issues—we are fully committed to resolving them before the discussion period ends. Your feedback means a great deal to us.

---

> ### Author Response · Authors · 2025-11-28
>
> Dear Reviewer, we sincerely appreciate the time and care you have devoted to evaluating our submission. With the discussion period concluding in a few days, we would like to gently inquire whether you have any remaining questions or if there are further analyses or experiments you would like us to explore. We are more than willing to provide additional theoretical insights or experimental evidence promptly to help clarify any aspects that may still be unclear. Your feedback is extremely important to the development of our work, SynHLMA, and we would be deeply grateful for your continued consideration.

---

### Official Review · Reviewer_V4CR · 2025-11-10

**Soundness:** 3
**Presentation:** 2
**Contribution:** 3
**Rating:** 6
**Confidence:** 2

**Summary:**

The authors propose SynHLMA, a framework for generating hand–articulated-object interaction sequences from a point cloud and natural-language instruction. The method (i) discretizes each manipulation frame using a multi-stage VQ-VAE into tokens representing the global hand pose (⟨g⟩), local articulation (⟨l⟩), refinement (⟨r⟩), and object joint (⟨j⟩); (ii) trains a manipulation language model to autoregressively predict these tokens conditioned on textual input; and (iii) introduces an articulation-aware loss that combines hand–object penetration, pose-consistency, and joint-configuration terms. The model supports three tasks—generation, prediction, and interpolation. The authors also propose HAOI-Lang, a simulated dataset containing approximately 50,000 manipulation sequences paired with GPT-4–generated captions across seven object categories (stapler, laptop, scissors, cabinet, dishwasher, eyeglasses, and box). Experiments demonstrate state-of-the-art performance across six quantitative metrics.

**Strengths:**

1. The author proposes a new human–articulated-object interaction framework by LoRA training an LLM.

2. The author also proposes a new hand–articulated-object dataset that includes natural language descriptions.

**Weaknesses:**

Weaknesses:

1. Previous work such as HOIGPT follows a paradigm that is quite similar to the proposed approach. The paper should provide more detailed comparisons with these methods. In particular, it would be helpful to clarify whether those baselines were trained on the same dataset as SynHLMA to ensure fairness and reproducibility.

2. In the ablation studies, please specify the version of LLaMA used. While the choice of LLaMA as a comparison baseline is reasonable, there are other state-of-the-art base models (e.g., Qwen, Gemma) that have shown superior ability in different aspects. Including results for these models would make the evaluation more comprehensive and up-to-date.

Minor Comments:

Line 087 — a space is missing before “By”.

Figures 2 and 3 are cluttered and difficult to read. Simplifying these figures and highlighting the key modules would make the workflow easier to understand.

Please clarify whether Figure 2 corresponds to the model trained with LLaMA or Vicuna.

Line 418 — the LaTeX equation formatting is broken and should be fixed for readability.

**Questions:**

See weaknesses.

---

> ### Author Response · Authors · 2025-11-18
> **Rebuttal by Authors**
>
> We sincerely appreciate your valuable comments. Below we provide a point-by-point response to your questions. If you have any additional concerns, please feel free to let us know. We genuinely look forward to further discussion with you.
>
> ---
>
> ### **1. Comparison with Other Methods**
>
> Thank you for your insightful suggestions. All baselines and our method **SynHLMA** are trained on the dataset proposed in this work, **HAOI-lang**.
>
> Compared with HOIGPT, our approach models joint states explicitly through the **Joint Prediction Decoder**, enabling higher-quality and more fine-grained static HAOI generation through the multi-stage VQ-VAE. Moreover, to ensure both physical plausibility in VQ-VAE reconstruction and temporal consistency in the manipulation language model, we introduce a series of **articulation-aware losses**. A standard VQ-VAE relying solely on reconstruction loss cannot guarantee grasp–joint consistency or accurate articulation dynamics.
>
> Specifically:
>
> - **Stage 1 (per-frame modeling):**
>   We apply **HAOI Penetration Loss** and **Joint Aware Loss** to ensure physical plausibility and correct articulation perception in each static HAOI frame.
>
> - **Stage 2 (manipulation sequence modeling):**
>   After the manipulation language model outputs the token sequence for each frame, the multi-stage VQ-VAE decodes hand and object rotation matrices. We compute the transition values between adjacent frames and enforce **temporal consistency** between hand pose and object articulation.
>
> To the best of our knowledge, this is the first work to introduce an articulation-aware loss
> into a discrete VQ-VAE framework for HAOI generation, explicitly enforcing joint-state
> consistency across both spatial and temporal dimensions.
>
> In Sec. 4.3 of the main paper, we have demonstrated the effectiveness of the articulation-aware loss through ablation studies. The results are shown below:
>
> #### **Ablation on Articulation-Aware Loss**
>
> | Setting | FID ↓ | IV ↓ | ADE ↓ |
> |---------|--------|---------|----------|
> | w/o $𝓛_{artic}$ | 15.872 | 6.452 | 1.041 |
> | w/o $𝓛_C$ | 14.990 | 6.104 | 1.112 |
> | w/o $𝓛_{artic}$ ∧ $𝓛_C$ | 16.840 | 6.987 | 1.233 |
>
> In the above table, we provide a detailed evaluation of the proposed articulation-aware loss designed for articulated objects. The experimental results show that:
>
> 1. Without the static-frame constraint term $𝓛_{artic}$, both IV and FID increase significantly, indicating reduced physical plausibility and contact consistency in HAOI grasping, as well as larger errors in the generated grasp poses.
> 2. When removing $𝓛_C$, the ADE rises markedly, suggesting that temporal inconsistencies—such as noticeable "jittering" or "discontinuities" in hand-object angular transitions—become more pronounced.
> 3. When both loss terms are ablated together, the overall sequence generation quality drops to the worst level.
>
> These results demonstrate that the proposed articulation-aware loss effectively links the key components of the model and explicitly encodes both static HAOI states and continuous manipulation dynamics, thereby ensuring high-quality generation.
>
> ---
>
> ### **2. Questions Regarding the Manipulation Language Model Backbone**
>
> Thank you for the constructive comments. We use **Vicuna-7B-v1.5** as the backbone. In Sec. 4.3 of the revised manuscript, we have added comparisons with **Qwen2.5-7B-Instruct** and **Gemma-7B-it**.
>
> Our findings are:
>
> - **Qwen** performs worse, possibly because its ability in fine-grained action description and numerical reasoning is slightly weaker. Another potential reason is that our training captions are all in **English**, while Qwen may be more optimized toward bilingual or Chinese-centric training corpora.
> - **Gemma** performs comparably to Vicuna, but since Gemma has a larger base model, due to limited computation resources we trained it with **4-bit quantization**, whereas Vicuna was trained with **8-bit precision**, leading to a slight performance gap.
>
> Additionally, we would like to clarify a point about Fig. 2:
> The figure uses **“LLaMA”** to represent that *any* LLaMA-based or LLaMA-finetuned LLM can be used in our framework. In practice, the actual model used in experiments is **Vicuna**, a finetuned variant of LLaMA. The results are shown below:
>
> #### **Ablation on Manipulation Language Model Setting**
>
> | Backbone | FID ↓ | MMDist ↓ | IV ↓ |
> |---------|--------|-----------|--------|
> | w/ Llama | 51.911 | 17.222 | 9.244 |
> | w/ Qwen | 73.590 | 25.272 | 6.848 |
> | w/ Gemma | 22.576 | 16.228 | 2.317 |
>
> ---

---

> ### Author Response · Authors · 2025-11-18
> **Rebuttal by Authors**
>
> ### **3. Writing and Presentation Issues**
>
> We greatly appreciate your careful reading and helpful suggestions. We have corrected all the issues you pointed out. Beyond that, we performed a thorough revision of the manuscript to ensure clarity and fluency.
>
> In Section 3.3, we have reorganized the presentation flow by first introducing the VQ-VAE responsible for joint prediction, followed by the multi-level VQ-VAE used for reconstructing HAOI grasping parameters. We then separately describe the first component of the articulation-aware loss and the loss functions used in the multi-level VQ-VAE. In Section 3.4, we provide a more detailed explanation of the articulation-aware loss.
>
> We have also refined Fig. 2 to address the occlusion issue related to the tensor $C_\mathcal{O}$. In addition, we include several new ablation studies (e.g., articulation-aware loss, comparisons with Qwen and Gemma, continuous vs. discrete feature representations), which together make the paper more comprehensive and convincing.
>
> We add an analysis of failure cases in the appendix to further enhance the completeness of the work.
>
> Finally, We corrected formatting errors throughout the manuscript.

---

> ### Author Response · Authors · 2025-11-24
>
> Dear reviewer, we sincerely appreciate your thoughtful and valuable comments. We have provided a point-by-point response, and we truly hope that our clarifications address your concerns. Please let us know if there are any remaining issues—we are fully committed to resolving them before the discussion period ends. Your feedback means a great deal to us.

---

> ### Author Response · Authors · 2025-11-28
>
> Dear Reviewer, we sincerely appreciate the time and care you have devoted to evaluating our submission. With the discussion period concluding in a few days, we would like to gently inquire whether you have any remaining questions or if there are further analyses or experiments you would like us to explore. We are more than willing to provide additional theoretical insights or experimental evidence promptly to help clarify any aspects that may still be unclear. Your feedback is extremely important to the development of our work, SynHLMA, and we would be deeply grateful for your continued consideration.

---

### Author Response · Authors · 2025-11-28
**Following up on our rebuttal and discussion**

Dear Area Chair and Reviewers,

Thank you sincerely for the time and care you have devoted to our submission. It has been more than two weeks since the discussion period began, and we have carefully addressed each of the reviewers’ comments with the utmost attention. We are truly grateful for your thoughtful insights and constructive suggestions, which have greatly helped us improve our work.

Below, we provide a brief summary of our main responses:

1. We elaborated on the rationale and novelty of our proposed articulation-aware loss, as well as the design of the Joint Prediction Decoder and the multi-level VQ-VAE architecture.
   (Please refer to our response to Reviewer nYnu, Point 3, and Reviewer ENDg, Point 2.)

2. We addressed issues regarding formatting and the logical flow of the manuscript.
   (Please see our response to Reviewer V4CR, Point 3, and Reviewer ENDg, Point 1.)

3. We provided more detailed comparisons with related works, such as HOIGPT and SemGrasp.
   (Please refer to our response to Reviewer V4CR, Point 1, and Reviewer 9XZF, Point 3.)

4. We included additional comparisons with diffusion-based HOI generation methods, such as NL2Contact.
   (Please see our response to Reviewer nYnu, Point 2.)

5. We discussed how different multimodal large language models, including Qwen and Gemma, influence the performance of our overall approach.
   (Please refer to our response to Reviewer V4CR, Point 2, and Reviewer nYnu, Point 5.)

6. We further explored how HAOI generated in simulation could be transferred to real-world scenarios.
   (Please see our response to Reviewer nYnu, Point 1, and Reviewer ENDg, Point 5.)

7. We examined whether our proposed method can adapt to new scenarios through few-shot learning.
(Please refer to our response to Reviewer nYnu, Point 8, and Reviewer ENDg, Point 4.)

8. We clarified the limitations of the articulation-aware loss, specifically that it cannot unify translational and rotational axes.
   (Please see our response to Reviewer nYnu, Point 6.)

9. We added more qualitative visualizations, including additional HAOI examples, videos, and analyses of several failure cases.
   (Please refer to our response to Reviewer ENDg, Point 3.)

10. We provided further details regarding the construction of our generated dataset, HAOI-Lang.
   (Please see our response to Reviewer 9XZF, Point 1, and Reviewer nYnu, Point 4.)

We sincerely hope that these clarifications and additional experiments adequately address your concerns. We remain fully committed to further discussion and are prepared to provide any additional information or follow-up responses as needed. Your updated insights would be greatly appreciated.

Sincerely,
Authors of Paper 6863

---

### Author Response · Authors · 2025-12-02

Dear Area Chair and Reviewers,

As part of a further careful self-review of our manuscript, we identified and corrected a number of issues that were not previously noticed, including minor spelling errors, ambiguous or potentially misleading descriptions, and several formatting and detail-level inconsistencies. All corresponding changes have been incorporated into the updated version (Dec 2 revision) and are highlighted in red for your reference.

---

### Meta-Review · Area_Chair_ahkv · 2026-01-06

**Summary:**

This paper proposes SynHLMA, a framework for synthesizing hand–language manipulation sequences for articulated objects using discrete representations and a manipulation language model. The work combines a multi-level VQ-VAE for discretizing hand–object interactions, an articulation-aware loss, and a language-conditioned autoregressive model trained on a newly constructed simulated dataset. The paper aims to support multiple tasks, including HAOI generation, prediction, and interpolation, and further demonstrates an application to robotic dexterous manipulation.

Reviewers acknowledged the substantial engineering effort, the scale of the dataset construction, and the comprehensive set of experiments and ablations. However, the overall consensus is that the paper suffers from significant issues in clarity, presentation, and positioning, which make it difficult to assess the true novelty and technical contribution. These issues, combined with concerns about conceptual rigor and over-claiming, ultimately outweigh the strengths of the empirical results.

**Reviewer Concerns:**

Concerns addressed by the rebuttal:
The rebuttal provides additional clarifications on implementation details, dataset construction, and loss formulations. Some reviewers’ questions regarding experimental settings and ablation interpretations were partially addressed, and the authors reiterated the motivation behind discrete representations and articulation-aware constraints.

Existing concerns:
Several major concerns remain unresolved. First, multiple reviewers pointed out that the presentation is weak and often confusing: the paper is difficult to follow, key concepts are introduced unclearly, and important design choices are insufficiently motivated. This significantly hinders the readability and scientific communication of the work, independent of its technical content.

Second, reviewers consistently noted that the novelty is overstated. Many components, such as VQ-VAE discretization, language-conditioned autoregressive modeling, articulation constraints, and simulator-based data generation, are largely adaptations or combinations of existing techniques. While the integration is non-trivial, the paper frequently frames these contributions as more fundamental or novel than they appear to be.

Third, there are conceptual concerns about the framing of the manipulation language model and the strength of the claims regarding semantic understanding and generalization. Some claims are not fully supported by the experiments, and the connection between the proposed representation and higher-level manipulation reasoning remains ambiguous.

Finally, although the experimental section is extensive, reviewers questioned whether the reported improvements sufficiently justify the added complexity of the framework, especially given the reliance on simulated data and LLM-generated annotations.

**Reviewer Scores:**

Reviewer V4CR: Likely to maintain their original score, as their primary concerns about clarity, over-claiming, and positioning were not fully resolved.

Reviewer 9XZF: Likely to maintain their score, given continued concerns about presentation quality and the difficulty of assessing novelty.

Reviewer nYnu: Might slightly adjust their score after discussion but would still remain on the negative side, as the rebuttal does not fundamentally address the conceptual and framing issues.

Reviewer ENDg: Likely to maintain their original score, viewing the work as technically heavy but insufficiently polished and positioned for acceptance.

---

### Decision · Program_Chairs · 2026-01-26

Reject